# Sync or Sink: Bounds on Algorithmic Collective Action with Noise and Multiple Groups

## Abstract

Collective action against algorithmic systems, which enables groups to promote their own interests, is poised to grow. Hence, there will be growth in the size and the number of distinct collectives. Currently, there is no formal analysis of how coordination challenges within a collective can impact downstream outcomes, or how multiple collectives may affect each other's success. In this work, we aim to provide guarantees on the success of collective action in the presence of both coordination noise and multiple groups. Our insight is that data generated by either multiple collectives or by coordination noise can be viewed as originating from multiple data distributions. Using this framing, we derive bounds on the success of collective action. We conduct experiments to study the effects of noise on collective action. We find that sufficiently high levels of noise can reduce the success of collective action. In certain scenarios, large noise can sink a collective success rate from $100\%$ to just under $60\%$. We identify potential trade-offs between collective size and coordination noise; for example, a collective that is twice as big but with four times more noise experiencing worse outcomes than the smaller, more coordinated one. This work highlights the importance of understanding nuanced dynamics of strategic behavior in algorithmic systems.

## 1 Introduction

As large platforms (social media, e-commerce) grow, groups within them find it increasingly important and necessary to act strategically (e.g., by changing ratings) in these platforms to get their desired outcome. From delivery drivers coordinating to earn higher wages [SHMD24], to fans promoting their favorite artist [XZF+25, Nug21], to protesting controversial businesses [Pay24], this type of strategic behavior is poised to grow. Strategic behavior ultimately affects the downstream data distributions used to train models. As the incentive to participate increases, we need a more nuanced understanding of the impacts on a system; this includes both understanding what happens with multiple distinct collectives acting on a system as well as considering how internal collective dynamics may impact outcomes. These factors will all impact the downstream data distribution. However, currently, there are no tools or analytical frameworks that incorporate these factors (coordination noise, multiple collectives) to assess their impact on the success of collective action.

In classical collective action problems related to managing common-pool resources, Ostrom notes that smaller, more homogeneous groups are more effective at overcoming barriers to collective action [Ost90]. She finds that a clear organizational structure and well-defined roles are crucial—one possible interpretation is that smaller groups can more readily and faithfully implement and coordinate their actions. An analogy in the computational world is "noise": unwanted modifications or deviations from intended behavior (e.g., signal processing). This "noise" can result from poor coordination and may reduce effectiveness.

Submitted to 39th Conference on Neural Information Processing Systems (NeurIPS 2025). Do not distribute.

In algorithmic settings, collective action is mediated through strategic data modifications to influence model behavior. These changes in data distribution can have significant downstream ramifications. [HMMDZ23] provided theoretical bounds on the success of collective action in the case of a single, unified collective. The collective modifies their data (e.g., by inserting a watermark), which is then used in training. The learning algorithm observes a distribution $\mathcal{P}$: a mixture of data induced by the collective's action ($\mathcal{P}_1$) and the underlying data distribution $\mathcal{P}_0$. These bounds help identify the conditions under which collective action may be more successful. However, this analysis assumes perfect coordination within the collective. [KVKS25] developed a framework to identify the factors affecting collective action involving multiple collectives, using simulations to illustrate several outcomes. However, they do not provide any theoretical bounds on success based on the data distributions induced by each collective. A formal understanding of how both noise and multiple collectives interact is key to understanding real-world algorithmic collective action.

In this work, we provide new guarantees on algorithmic collective action in both the presence of noise and multiple distinct collectives. We do this by introducing *multiple distributions* that feed the ultimate observed data distribution $\mathcal{P}$. Instead of considering just a mixture of $\mathcal{P}_0$ and $\mathcal{P}_1$, we study how the addition of $\mathcal{P}_2$ affects a collective's success. Importantly, the source of these distributions can vary: it could occur from multiple distinct collectives or because of coordination challenges within a single collective inducing a new, second data distribution.

A collective that is highly coordinated is one where the size of $\mathcal{P}_2$ is small and similar to $\mathcal{P}_1$, while ones with less cohesion have a larger gap between $\mathcal{P}_1$ and $\mathcal{P}_2$. Multiple collectives can be analyzed as behavior coming from two distinct distributions. Our contributions are as follows:

**Theoretical bounds with multiple distributions:** We are the first to establish the lower bounds for the success of collective action in the presence of multiple distributions. Prior work has either only analyzed a single distribution coming from a single collective or provided empirical outcomes with multiple collectives [KVKS25, HMMDZ23]. We are able to relate this bound to the similarity of the distributions and collective size. This approach can effectively handle many different scenarios: including when there's a fraction of a collective performing actions imperfectly or when multiple distinct collectives are acting upon a system.

**Empirical impact of noise:** We study the impact of noise in collective action. Prior work [HMMDZ23] has only shown outcomes with perfectly coordinated collectives. We find that different types of noise can impact the success of collective action and find in some situations, smaller, less noisy groups perform better than larger groups with more noise.

We first present a formulation of the algorithmic collective action problem. We establish bounds on the success of collective action with multiple distributions. Our experiments extend [HMMDZ23] by considering a text classification task with noise. We find scenarios where noise sinks success a task from nearly $100\%$ without noise to under $60\%$. We discuss possible trade-offs in size and noise where small groups with less noise ($0.25\%$ collective size with $10\%$ noise) outperform larger ones with more noise ($0.5\%$ with $40\%$ noise), which can inform organizers of collective action.

## 2 Related Work

Collective action against algorithms has been documented in contexts such as ridesharing [Lei21, WAC21, WVM21, JGV21, Has20] and in data campaigns aimed at promoting pro-social outcomes [MF22, VH21, VLT+21, WED22]. [SHMD24] provides a specific computational model for the delivery driver case, which can be used to analyze incentives and outcomes in these scenarios. [XZF+25, XFS+25] examine some of the real-world structures involved in promoting online collective action, such as what motivates certain users to act as organizers. They also discuss practical challenges in influencing online marketplaces, including coordinating across different types of devices and teaching users how to engage effectively with the platform to achieve a specific goal. The concept of noise has been used to simulate communication challenges in agent-based modeling [WSLA13, ASJCB02, BBG23] and in modeling bounded rationality [Kah03, Con96, Qui07]. However, it has not been studied in the context of online collective action.

Other work has examined collective action in algorithmic settings. [HMMDZ23] first examines how different types of strategies can enable small collectives to have a substantial impact on classification outcomes. [BMD24] extends this analysis to examples involving recommender sys-

tems. [BDFSS24] examines how the specifics of the learning algorithm impact collective action. [KVKS25] empirically analyzes the interaction between two collectives. None of these works, however, offer formal guarantees for successful collective action involving multiple data distributions. Our work offers a more comprehensive theoretical understanding of realistic strategic dynamics.

# 3    Problem Formulation

Here we first will describe collective action as described by [HMMDZ23]. We will then extend this scenario to include a second distribution. For convenience, Appendix A summarizes the notation.

## 3.1    Algorithmic Collective Action with One Distribution

We consider the case where a firm wishes to deploy a classifier $f$ trained on some data. Let $f : \mathcal{X} \to \mathcal{Y}$. We define a classifier to be $\epsilon_1$ suboptimal under a distribution $P_1$ if there exists a $P'$ with $TV(P_1, P') \leq \epsilon_1$ such that $f = \arg\max_{y \in \mathcal{Y}} P'(y|x)$ where TV is the total variation distance.

The firm's goal is to minimize the loss with respect to some objective function. The data they train on exists in $\mathcal{Z} = \mathcal{X} \times \mathcal{Y}$ representing the features $\mathcal{X}$ and labels $\mathcal{Y}$. For our purposes, we can think of every data point $z \in \mathcal{Z}$ as belonging to an individual. A collective wishes to work together in order to create a specific outcome on this classifier for a certain set of inputs which we formalize below.

Consider the base distribution $\mathcal{P}_0$ on $\mathcal{Z}$. We define $\mathcal{P}_1$ to be the distribution induced by the collective's intended action. This is operationalized by strategy, $h_1 : \mathcal{Z} \to \mathcal{Z}$ where $h_1 \in \mathcal{H}$ the set of all available strategies. Examples of potential strategies may include watching a video for a certain amount of time, giving a specific review score, etc. In implementing this strategy, the collective wishes to create an association between the specific inputs to a target label $y^*$.

The collective plants a "signal", which is done via function $g_1 : \mathcal{X} \to \mathcal{X}$ which takes some original input $x \sim \mathcal{P}_0$ and modifies it. Examples may include adding certain words to text, adding a watermark to a video, or reviewing an extra product on a marketplace. This planted signal is intended to create the association between the set of inputs generated by $g_1(x)$ and a target label $y^*$.

In some situations, it may also be possible for the collective to act on both the input data ($\mathcal{X}$) and potentially the output label ($\mathcal{Y}$). For the *feature-label* strategy, members can each change their feature and label. For the *feature-only* strategy, they can only change their features. These strategies, using $h_1$ and signal $g_1$, induce the collective's distribution $\mathcal{P}_1$. Let $\alpha_1$ be the fraction of users participating in the collective. We can write the distribution seen by the learning algorithm as a mixture of the distribution of the collective's data and the unmodified data.

$$\mathcal{P} = \alpha_1 \mathcal{P}_1 + (1 - \alpha_1)\mathcal{P}_0$$

Where $\mathcal{P}_1$ represents the distribution of $h_1(z)$ where $z \sim \mathcal{P}_0$. The collective's objective is to associate the signal $g_1$ with the target $y^*$. The success rate can be written as

$$S(\alpha_1) = \Pr_{x \sim \mathcal{P}_0}[f(g_1(x)) = y^*]$$

Success also depends on how the signals generated by the collective conflict or compete with data present in the underlying distribution. Let $\mathcal{X}_1 = \{g_1(x) : x \in \mathcal{X}\}$ the signal set. We define this as suboptimality gap of $\mathcal{X}_1$ on distribution $\mathcal{P}_0$ for a target $y^*$ as

$$\Delta_1^0(y^*) = \max_{x \in \mathcal{X}_1}(\max_{y \in \mathcal{Y}} \mathcal{P}_0(y|x) - \mathcal{P}_0(y^*|x))$$

We also use $\mathcal{P}_0(\mathcal{X}_1)$ to refer to the background overlap of signal set $\mathcal{X}_1$ - intuitively, this represents how common the modified input $\mathcal{X}_1$ is in the background distribution $\mathcal{P}_0$ Based on these definitions, [HMMDZ23] finds a lower bound on the success for this classifier in the single, perfectly coordinated collective for both the *feature-label* strategy and the *feature-only* strategy, restated here.

**Theorem 1** (Feature-label). *[HMMDZ23] The success rate of a collective with size $\alpha_1$ on a $\epsilon_1$ classifier with target $y_1^*$ with signal set $\mathcal{X}_1$ is bounded below by*

$$S(\alpha_1) \geq 1 - \frac{1 - \alpha_1}{\alpha_1} \mathcal{P}_0(\mathcal{X}_1) \frac{(1 - \epsilon_1)\Delta_1^0(y_1^*) + \epsilon_1}{1 - 2\epsilon_1}$$

**Theorem 2** (Feature-only). *[HMMDZ23] The success rate of a collective with size $\alpha_1$ on a $\epsilon_1$-suboptimal classifier with target $y_1^*$ with distribution $\mathcal{X}_1$ where there exists a $p$ such that $\mathcal{P}_0(y^*|x) \geq p, \forall x \in \mathcal{X}$ is bounded below by*

$$S(\alpha_1) \geq 1 - \frac{1 - (1 - \epsilon_1)p - \epsilon_1\alpha}{(1 - \epsilon_1)p\alpha - \epsilon_1\alpha}\mathcal{P}_0(\mathcal{X}_1)$$

However, these theorems do not consider a second distribution. We first develop our multiple distribution setup, which we will use to expand beyond the above bounds.

## 3.2 Collective Action with Multiple Distributions

Here, we extend beyond [HMMDZ23] by considering another data distribution. This may arise from internal coordination challenges or a separate collective.

We consider distribution $\mathcal{P}_2$ induced by a strategy $h_2$. $h_2$ can be completely distinct from $h_1$ or can represent a noisy version of $h_1$ (e.g $h_2(z) = h_1(z) + \delta$). $\mathcal{P}_2$ is the distribution of $h_2(z), z \sim \mathcal{P}_0$. For example, suppose a collective aims to create an association between a word $A$ existing at the top of a document and being rated as an important user $y^*$. Members could mistakenly use a similar word $B$ instead of $A$ or place $A$ at the end of the text instead – this would be represented by $\mathcal{P}_2$.

We can write the final distribution observed by the learning model as

$$\mathcal{P} = \alpha_1\mathcal{P}_1 + \alpha_2\mathcal{P}_2 + (1 - \alpha)\mathcal{P}_0$$

where $\alpha_1$ represents the fraction of people who implemented collective one's strategy, $\alpha_2$ represents a separate group's interactions. This can arise because the correct "action" was not properly shared with all group members, people getting confused about what exactly to do and not executing faithfully, or potential infiltrators within the group. We let $\alpha = \alpha_1 + \alpha_2$ is the total fraction of people attempting to implement some strategy. We can also equivalently consider $r = \frac{\alpha_1}{\alpha_1 + \alpha_2} = \frac{\alpha_1}{\alpha}$ as the fraction of people implementing the correct strategy.

Suppose a collective that wants to plant a single $g(x)$ where $x \in \mathcal{X}$ Let the signal set $\mathcal{X}_1 = \{g_1(x) : x \in \mathcal{X}\}$. Another group implements another signal, which may be a noisy version of $g_1$. Let $g_2$ represent the set of signals produced by the second group. $\mathcal{X}_2 = \{g_2(x) : x \in \mathcal{X}\}$.

The presence of another distribution requires us to expand upon our definition of suboptimality to consider, across any two distributions, how often the signals from set $\mathcal{X}_i$ are present on the distribution $P_j$. Our expanded definition of suboptimality is:

$$\Delta_i^j(y^*) = \max_{x \in \mathcal{X}_i^*} (\max_{y \in \mathcal{Y}} \mathcal{P}_j(y|x) - \mathcal{P}_j(y^*|x))$$

Intuitively, this measures how "confusing" signals from $\mathcal{X}_i$ looks to $\mathcal{P}_j$ when targeting $y^*$.

# 4 Multiple Distributions and Success of Collective Action

Based on our expanded definitions, we can quantify the impact that multiple distributions have on the effectiveness of a collective. We consider both the feature-label strategy and the feature-only strategy. Both are important to study, as in different contexts, collectives may be able to alter both their input features and labels, or just their features. We state the theorems for two distributions here and defer the proofs and generalization to $n$ distributions to the Appendix E. While earlier work focused on a single, perfectly coordinated collective, we quantify how either internal disruptions or the presence of a second collective affects the original group's success. Conceptually, the proofs follow by considering the new mixture distribution and the interaction between the new distributions.

For the feature-label strategy, we have the following result (see Appendix C for full proof):

**Theorem 3** (Feature-label with two distributions). *Consider distribution $\mathcal{P}_1$ and $\mathcal{P}_2$ which are distributed according to $h_1(x)$ and $h_2(x)$ respectively, where $x \sim \mathcal{P}_0$. Let $y_1^*$ be the target class. Then success for the first collective against an $\epsilon_1$ classifier to be lower bounded by*

$$S(\alpha_1) \geq 1 - \frac{\alpha_2}{\alpha_1}\mathcal{P}_2(\mathcal{X}_1)\frac{(1 - \epsilon_1)\Delta_1^2(y_1^*) + \epsilon_1}{1 - 2\epsilon_1} - \frac{1 - \alpha}{\alpha_1}\mathcal{P}_0(\mathcal{X}_1)\frac{(1 - \epsilon_1)\Delta_1^0(y_1^*) + \epsilon_1}{1 - 2\epsilon_1}$$

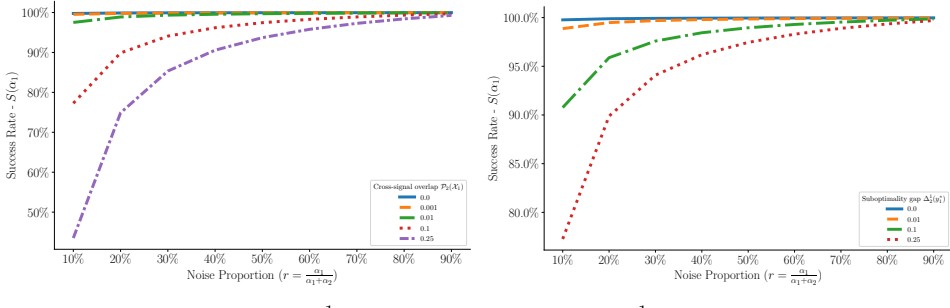

(a) Varying $\mathcal{P}_2(\mathcal{X}_1)$ for fixed $\Delta_2^1(y_1^*) = 0.25$  (b) Varying $\Delta_2^1(y_1^*)$ for fixed $\mathcal{P}_2(\mathcal{X}_1) = 0.25$

Figure 1: Relationship between $\Delta_2^1(y_1^*)$, $\mathcal{P}_2(\mathcal{X}_1)$ and success bound. We fix $\alpha_1 + \alpha_2 = 0.3$ and vary the proportion $r$. We fix $\epsilon_1 = 0$, $\Delta_0^1 = 0.01$, and $\mathcal{P}_0(\mathcal{X}_1) = 0.01$

Compared with Theorem 1, Theorem 3 introduces another term that describes the relationship between $\mathcal{P}_1$ and $\mathcal{P}_2$ independently of the relationship between $\mathcal{P}_0$ and $\mathcal{P}_1$. It consists of the cross-signal overlap between signal set $\mathcal{X}_1$ on distribution $\mathcal{P}_2$ the suboptimality gap of the target $y_1^*$ of signal set $X_1$ and $\mathcal{P}_2$. ($\Delta_1^2(y_1^*)$) and the relative sizes of said groups $\frac{\alpha_2}{\alpha_1}$. Figure 1 illustrates these relationships.

If we consider the distributions as coming from two collectives with differing objectives, we can also write down the success bound for the second collective as well.

**Corollary 3.1.** *If we consider $\mathcal{P}_2, \mathcal{X}_2$ as coming from a 2nd collective with (potentially) distinct objective, we can write the probability of success as*

$$S(\alpha_2) \geq 1 - \frac{\alpha_1}{\alpha_2}\mathcal{P}_1(\mathcal{X}_2) \cdot \frac{(1-\epsilon_2)\Delta_2^1(y_2^*) + \epsilon_2}{1 - 2\epsilon_2} - \frac{1-\alpha}{\alpha_2}\mathcal{P}_0(\mathcal{X}_2)\frac{(1-\epsilon_2)\Delta_2^0(y_2^*) + \epsilon_2}{1 - 2\epsilon_2}$$

For the feature-only strategy we have the following result (Appendix D for full proof)

**Theorem 4** (Feature-only with two distributions). *Consider distribution $\mathcal{P}_1$ and $\mathcal{P}_2$ which are distributed by $h_1(x)$ and $h_2(x)$ respectively, where $x \sim \mathcal{P}_0$. Suppose there exist a $p$ such that $\mathcal{P}_0(y^*|x) \geq p, \forall x \in \mathcal{X}$. Then success for the first collective against an $\epsilon_1$ classifier (against $P_1^*$) is lower bounded by*

$$S(\alpha_1) \geq 1 - \frac{\alpha_2}{\alpha_1} \cdot \frac{\mathcal{P}_2(X_1^*) \cdot \Delta_1^2(y^*)(1-\epsilon_1)}{p(1-\epsilon_1) - \epsilon_1} - \frac{1-\alpha}{\alpha}\mathcal{P}_0(\mathcal{X}_1) \cdot \frac{(1-p)(1-\epsilon_1) + \epsilon_1}{p(1-\epsilon_1) - \epsilon_1}$$

In this bound we see the same $\mathcal{P}_2(\mathcal{X}_1)\Delta_2^1(y_1^*)\alpha_2(1-\epsilon_1)$ term which captures the cross-signal overlap, suboptimality gap and collective size.

These theorems illustrate how and to what extent the presence of a second distribution can hinder the first collective. If we consider the second distribution to be a noisy variation of the first one, this helps relate the noise characteristics to the success rate. For a second collective, these bounds provide insights into which scenarios collectives may be simultaneously successful or hindering each other. We examine these implications in Section 7.1.

## 5 Experimental Setup

To demonstrate the empirical implications, we study the case of noisy collective action. We extend [HMMDZ23] experiment on resume classification. We use the resume dataset introduced by [JT21] to finetune a BERT-based model for a multilabel prediction task. The goal for the classifier is to predict which set of careers someone is suited for based on the text in the resume. Members of the collective intend to plant a signal in their resume to get the resume classified to some target class $y^*$ inserting a specific character in a certain pattern. The intended signal in this case is to place this specific character every 20 words (full details in Appendix B).

We vary $r$, the proportion of users who perform an imperfect collective action by performing a noisy variation. Specifically, we consider the following types of noise variations.

**Correct Character Usage** The collective attempts to place a specific character into the resume. A noisy variation involves using a different character. We consider variations where the wrong character is sampled across a small subset (Random-Subset), as well sampled randomly across all possible characters (Random-Full).

**Modification Location** The collective intended action to place their character every 20 words. A noisy variation places the character in arbitrary locations (Displaced).

These choices are motivated by considering "benign" ways a group of people may misinterpret instructions. If the intended instruction is "Place character 'A' every 20 words", some users may focus more on the 20 words part and not use the right character, or some users may focus on using "A" and not place it every 20 words. We define the full set of variations in Table 1.

We also consider the context in which the noise occurs.

**Noised Input** For feature-label strategy, it is possible that the noise only gets applied in the input feature or gets applied to the feature as well as the label.

**Target's Underlying Frequency** The target class underlying frequency (in $\mathcal{P}_0$) has shown to play some impact in algorithmic collective action, especially with the feature only strategy [HMMDZ23]. We test a high frequency label (Software Developer) as well as a low frequency label (Database Administrator) as the target class.

Based on these factors, we ask the following questions.

**RQ1:** Does the frequency of noise (larger $r$'s) affect success more than noise variation?

**RQ2:** What class of strategies are more sensitive to noisy behaviors: feature-label or feature-only?

**RQ3:** How does the background frequency ($\mathcal{P}_0$)of the target class affect sensitivity to noise?

| Variation Name | Variation Description |
|---|---|
| Baseline | All members use the same characters and place every 20 words |
| Random-Subset | Some members insert a different character than the signal character, chosen from a small, fixed subset, placed every 20 words. |
| Random -Full | Some members insert a different character than the signal character, chosen from a large, fixed subset, placed every 20 words. |
| Displaced-Original | Some members place the collective's signal character at arbitrary places in the text |
| Displaced-Full | Some members place insert a randomly picked character from a large subset at arbitrary places in the text. |

Table 1: List of variations for noise deviations

# 6 Results

We finetune `distilbert-based-uncased` [SDCW19] for five epochs with default hyperparameters using Hugging Face transformer library [WDS$^+$20]. For the feature-label strategies, we vary $\alpha$, the percent of users in the collective from $0 - 1\%$. For the feature only strategies, we vary $\alpha$'s from $[0\%, 50\%]$.We consider noise rates $r$ from $[0\%, 50\%]$ (hence $\alpha_1 = r\alpha$). Noise variations for input text is detailed in Table 1. For noising labels, the label is changed uniformly at random across all possible labels. We measure success by calculating how often the target class ($y_1^*$) is predicted for text that has the desired modification $x^*$ in the test set. All figures are shown with a one standard deviation region shaded.

## 6.1 RQ1: Different Noise Variations

Here we consider the several types of noise variations (as described in Table 1). As a reminder, these different noise variations change how the different members of the collective may incorrectly

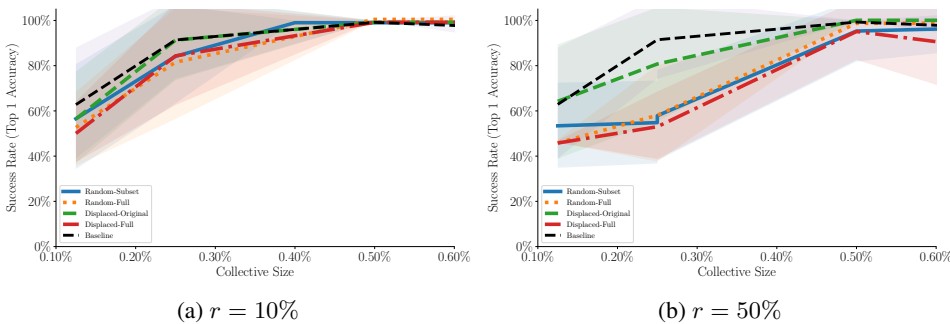

(a) $r = 10\%$            (b) $r = 50\%$

Figure 2: Comparison of different noise variations for different collective sizes. The x-axis here is the total collective size, measured by the percentage of the total population is participating. The $y$-axis is the success rate of causing resumes with the "true" signal to be classified to the target class. The black dashed line represents success rate without noise. The strategy is feature-label where noise is only applied to the features. We observe that making more characters for mistakes leads to lower success (Random-Subset vs Random-Full and Displaced-Original vs Displaced-Full)

implement a strategy. The $x$-axis represents the collective size while the $y$-axis show the collective's success criteria. Figure 2 shows how different noise variations result in different efficacies, both for low levels of noise (Figure 2a) and higher levels (Figure 2b). We find that different levels of noise have more of an impact when the collective size is relatively small; for sufficiently high levels of participation, all variants perform similarly and comparable to the baseline even with noise. At this lower levels, there is a sensitivity to the set of "wrong" characters to choose from: the Random-Subset strategy performs better than the Random-Full strategy. We see displacement while keeping the original character performs better than the Random-Subset - which keeps the placement consistent. For this specific model, this may imply that it is more sensitive to character coordination than the specific placement. We also find that across the different noise types, the size of the collective seems can overcome noise.

## 6.2    RQ2: Feature-Label vs Feature-Only

Here we examine how noise affects success for the feature-label vs feature-only strategy. We fix a level of participation and examine how success changes as noise increases. Figure 3 shows the decline in success for feature-label and feature-only strategies as a function of the percentage of the collective subject to noise. We examine three cases, a feature-label scenario where noise is applied to both the feature and the label (Figure 3a), a feature-label scenario where noise is applied to just the feature (Figure 3b) and the feature-only scenario (Figure 3c).

In Figure 3a we see with moderate levels of participation (0.5%) that noise has a major impact (from nearly 100% success rate to below 60%). Even at higher levels of participation (1%) we see, at the high end, noise impacting success. However, in Figure 3b, we see relative robustness to noise at the higher levels of participation (0.5% and above). The impact of noise in the feature-only strategy is a more gradual decline (comparisons are not apples to apples because the feature-only strategy requires higher participation levels for comparable success). Overall, noise that affects labels has a more significant impact on a collective's success.

## 6.3    RQ3: Impact of Baseline Frequency

Here, we examine how noise impacts target classes that appear more frequently in the $\mathcal{P}_0$ distribution compared to less commonly seen ones. Figure 4 shows the impact for feature-label strategies. We find that, counterintuitively, at higher levels of participation (e.g 0.25% and above) low baseline classes are less susceptible to noise. The opposite holds true in Figure 5, where the high frequency target class sees some change (some positive/some negative) with the presence of noise, the low frequency target class is mostly negatively affected by noise. This could be since, unlike in the feature-label strategy, the feature-only strategy cannot boost the presence of a target class in the training data ; diluting the signal with noise may more directly impact success.

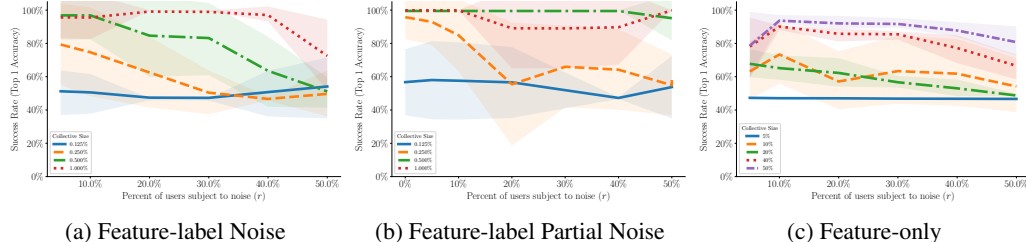

(a) Feature-label Noise     (b) Feature-label Partial Noise     (c) Feature-only

Figure 3: Impact of noise (Random-Subset) on different strategies. We consider various levels of noise for differing levels of participation. Each curve represents a fixed level of participation. The x-axis is the percentage of that group subject to noise. The left represents the feature-label strategy subject to noise on both the features and the labels. The middle represents the feature-label strategy where only the features are subject to noise. The right figure shows the feature-only strategy. We find that when noise affects both the feature and label, the decline in success is significant - especially at lower levels of participation. When noise just affects the inputs, the declines are more modest.

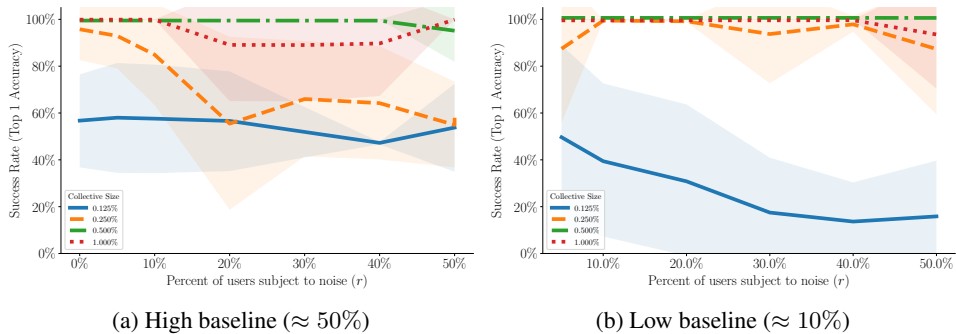

(a) High baseline ($\approx 50\%$)        (b) Low baseline ($\approx 10\%$)

Figure 4: Interaction between underlying frequency and susceptibility to noise (Random-subset) for feature-label strategy. Here we see for moderate levels of participation ($0.25\%$ and above), the low baseline scenario is more robust to noise.

# 7 Discussion

## 7.1 Analysis on Multiple Collectives

[KVKS25] examined how two distinct collectives with different objectives impact each other's efficacy – in short, when the same signal set is being used to target different classes, both collective's success rate is greatly reduced. This can be explained by the suboptimality gap ($\Delta_1^2(y_1^*)$) and the cross-signal overlap ($\mathcal{P}_2(\mathcal{X}_1)$). This overlap will be high since the two groups use the same signal. Because they are targeting two different classes, the suboptimality gap may also be large. They also find a case where two collectives, with different target classes and different character usage, still sinks both of their success rates. This can also be explained by the cross-signal overlap - if these character modifications look sufficiently "close" to each other, this term may be large and cause conflicts.

## 7.2 Trade-offs on Size and Noise

As strategic behavior on algorithmic systems continues to grow, understanding how deviations from a single, unified collective impact on a group's objective is crucial for both organizers of collective action and system developers. [XZF+25] observes the heterogeneity of members of a fan collective in terms of the actions available (e.g based on device type), as well as the detailed instructions they must give to others, which provides room for misinterpretation. This is also similar to parts of a collective action framework mentioned in [KVKS25] regarding the action availability and collective construction (i.e how the group is organized).

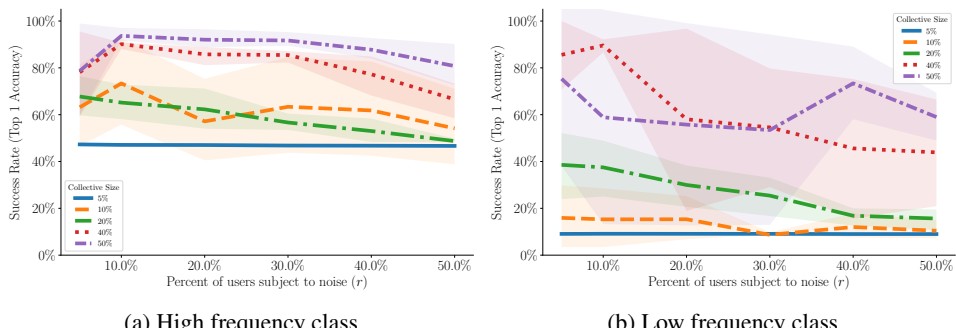

(a) High frequency class         (b) Low frequency class

Figure 5: Impact of noise (Random-subset) on the feature-only strategy. Compared to the feature-label strategy, both the high and low frequencies cases are impacted by noise across differing participation levels. The high frequency case, however, sees more gradual declines in the success rate compared to the low frequency case.

This theoretical analysis can also be used to help define how and where organizers can place resources. In particular, organizers can try to determine whether increasing the size of the collective ($\alpha_1$) is worth the trade-off in potentially increasing cross signal overlap, $\mathcal{P}_2(\mathcal{X}_1)$, or suboptimality gap, $\Delta_1^2(y_1^*)$. This trade-off has a traditional analogy in managing common pool resources [Ost90]: the importance of smaller, homogeneous groups to overcome barriers in collective action. We see it is possible that a smaller group with less noise may be able to outperform a larger group. In Figure 3a we observe a collective of size $0.25\%$ with $10\%$ noise rate outperforms a collective of size $0.5\%$ at $40\%$ noise rate. Noise here can be used to characterize a group's coordination efficiency. Different types of collectives and algorithmic systems might exhibit trade-offs between size and minimizing noise (if such a trade-off exists). For organizers of collective action knowledge of these characteristics can help to decide whether resources should be allocated to expand the total collective size or to more tightly coordinate within a small group.

### 7.3 Broader Impacts

Algorithmic collective action can generally be used when there's a difference between the goals of those who generate data and are affected by models and those who train and deploy them. Our aim is to continue the work on promoting socially valuable use cases of collective action ([FS22, AHJ+22]); however, we recognize that there could be malicious use cases. We believe that model developers should be aware of the strategic behavior that can lead to different data distributions.

### 7.4 Limitations and Future Work

We assumed that we could easily segregate between distributions. In some cases, this may be more natural (multiple collectives) while in others (benign noise) may be difficult to do in practice. We also considered a limited set of "noise" variations on a small set of data. Future work could explore more ecologically motivated types of deviations, including non-independent noise and adversarial actions. We also only considered a small classification task, different types of learning paradigms may be affected by noise differently – this would be an important avenue for future work.

## 8 Conclusion

We investigated the role of multiple distributions on the success of collective action. We derived lower bounds on the success rate in relation to the cross-signal overlap and suboptimality gap. We empirically evaluated noise in collective action. We found that noise variation matters, and that noise that affects labels more severally impacts collective action. We note that, for organizers, understanding the trade-offs between group size and the potential noise is important; different systems and types of actions may push organizers to invest more in effective coordination in a small group vs expansion. As strategic interest on algorithmic systems grows, both developers of algorithms and organizers of collective action must be aware of the potential that differing distributions has on system outcomes.

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

## A  Notation Table

We use considerable notation for defining relationship between collectives. Here we provide a concise table for referencing these symbols.

| Symbol | Description |
|--------|-------------|
| $TV(\cdot, \cdot)$ | Total variation distance between two distributions |
| $\epsilon_1$ | Suboptimality of the classifier |
| $f$ | Classifier trained on the mixture distribution |
| $\mathcal{X}$ | The entire feature space |
| $\mathcal{Y}$ | The set of labels |
| $\mathcal{Z}$ | The space of all training data $\mathcal{X} \times \mathcal{Y}$ |
| $\mathcal{P}$ | Observed mixture distribution |
| $\mathcal{P}_0$ | Base (non-strategic) data distribution |
| $\mathcal{P}_1$ | Distribution induced by collective one's intended strategy |
| $\mathcal{P}_2$ | Distribution induced by noise or a second collective's behavior |
| $h_1, h_2$ | Strategies applied by collectives or noisy actors |
| $g_1, g_2$ | Signal planting functions that modify the input $x$ |
| $\alpha_1, \alpha_2$ | Fraction of population following strategy $h_1, h_2$ respectively |
| $\alpha$ | Total participating fraction: $\alpha = \alpha_1 + \alpha_2$ |
| $r$ | Proportion of correctly aligned members: $r = \frac{\alpha_1}{\alpha}$ |
| $\mathcal{X}_1, \mathcal{X}_2$ | Signal set induced by $g$ (e.g $\mathcal{X}_1 = \{g_1(x) \mid x \in \mathcal{X}\}$) |
| $\mathcal{P}_i(\mathcal{X}_j)$ | Cross-signal overlap of $P_i$ on signal set $X_j$ |
| $\Delta_i^j(y^*)$ | Suboptimality gap for $X_i$ under $P_j$ |
| $S(\alpha_1)$ | Success probability for collective one |
| $y_1^*$ | Target class label for collective one |

Table 2: Summary of notation used in the paper.

## B  Experimental Details

As described in the main body we finetune a `distilbert-based-uncased` [SDCW19] for five epochs with default hyperparameters using Hugging Face transformer library [WDS+20]. This experimental setting is the same as [HMMDZ23, KVKS25]. The resume dataset from [JT21] was split into 20,000 training points and 5,000 test points. The default or baseline strategy was to place a specific character (in this case the '{' character) every 20 words. We evaluated on targeting the 'Software Developer' class, except for the 'low frequency class' where the target was 'Database Administrator.'

During training, each training point is select with a probability $r$ (the noise fraction) to be affected by a specific noise condition. If the point was selected to be noised, the noise would be applied to the text, and, for certain contains, the label as well.

**Random-Subset:** Instead of the intended character, a character from the list [U+2E18, '!', '?', '·'] was selected uniformly at random to be used.

**Random-Full:** Same as above, but the character chosen was from the set of all single characters in the vocabulary of the tokenizer.

**Displaced-Original:** Keep the intended character, but instead of placing every 20 words, choose uniformly at random (5-30 instances) places to insert the character.

**Displaced-Full:** Combine the character selection of Random-Full with the placement behavior of Displaced-Original.

413 To evaluate whether the signal was planted successfully, for each data point in the test set, we applied
414 the "true" signal and evaluated whether the trained classifier would predict the target class. We used
415 the top-one accuracy as done in [HMMDZ23] as the metric of success.

416 All experimental conditions were run 15 times. Experiments were run on a ppc64le based cluster
417 with V100 Nvidia GPUs. Each iteration took $30 - 40$ minutes to complete.

## C  Proof of Theorem 3

419 We restate the theorem here:

**Theorem.** *Consider distribution $\mathcal{P}_1$ and $\mathcal{P}_2$ which are distributed according to $h_1(x)$ and $h_2(x)$ respectively, where $x \sim \mathcal{P}_0$. Let $y_1^*$ be the target class. Then success for the first collective against an $\epsilon_1$ classifier to be lower bounded by*

$$S(\alpha_1) \geq 1 - \frac{\alpha_2}{\alpha_1}\mathcal{P}_2(\mathcal{X}_1)\frac{(1-\epsilon_1)\Delta_1^2(y_1^*) + \epsilon_1}{1 - 2\epsilon_1} - \frac{1-\alpha}{\alpha_1}\mathcal{P}_0(\mathcal{X}_1)\frac{(1-\epsilon_1)\Delta_1^0(y_1^*) + \epsilon_1}{1 - 2\epsilon_1}$$

420 *Proof.* We follow the same proof strategy as [HMMDZ23].

421 We consider the multiple distributions present the overall data distribution $\mathcal{P}$ $\mathcal{P}_0$ base distribution;
422 $\mathcal{P}_1$ group 1's distribution; $\mathcal{P}_2$ group 2's distribution

423 We write the mixture distribution as :

$$\mathcal{P}(x, y_1^*) = \alpha_1 \mathcal{P}_1(x, y_1^*) + \alpha_2 \mathcal{P}_2(x, y_1^*) + (1-\alpha)\mathcal{P}_0(x, y_1^*) \tag{1}$$

424 We also define the suboptimality gap on distribution $i$ for signal set $j$ and target label $y^*$

$$\Delta_i^j(y^*) = \max_{x \in X_i^*}(\max_{y \in \mathcal{Y}} \mathcal{P}_j(y|x) - \mathcal{P}_j(y^*|x)) \tag{2}$$

425 We define the suboptimality gap for a specific point $x$ as

$$\Delta_{i,x}^j(y^*) = \max_{y \in \mathcal{Y}} \mathcal{P}_j(y|x) - \mathcal{P}_j(y^*|x) \tag{3}$$

426 Our goal is to find, for which value of $\alpha_1$ can we guarantee the model to classify a point $x \in \mathcal{X}_1$ to
427 the target class $y_1^*$

428 First consider an $\epsilon = 0$ classifier. If the model $f$ classifies any point $x \in \mathcal{X}$ to $y_1^*$, it must mean that
429 $\mathcal{P}(y_1^*|x) > \mathcal{P}(y_1|x)$ or equivalently $\mathcal{P}(x, y_1^*) - \mathcal{P}(x, y_1) > 0$ for every $y_1 \neq y_1^*$.

430 In this strategy, both the features and labels are changed. Since $\mathcal{P}_1$ is the distribution which correctly
431 performs the intended collective action, every point $x_1^* \in \mathcal{X}_1$ maps to $y_1^*$. Therefore we can simplify
432 and get

$$\mathcal{P}(x, y_1^*) = \alpha_1 \mathcal{P}_1(x) + \alpha_2 \mathcal{P}_2(x, y_1^*) + (1-\alpha)\mathcal{P}_0(x, y_1^*)$$

433 Now, for $y \neq y_1^*$ we have

$$\mathcal{P}(x, y) = \alpha_1 \mathcal{P}_1(x, y) + \alpha_2 \mathcal{P}_2(x, y) + (1-\alpha)\mathcal{P}_0(x, y)$$

434 Given $\mathcal{P}_1$ maps everything to $y_1^*$ and nothing else this first term is 0 so we can write simplify to

$$\mathcal{P}(x, y) = \alpha_2 \mathcal{P}_2(x, y) + (1-\alpha)\mathcal{P}_0(x, y)$$

435 So we can write $\mathcal{P}(x, y_1^*) - \mathcal{P}(x, y) > 0$ as

$$\alpha_1 \mathcal{P}_1(x) + \alpha_2 \mathcal{P}_2(x, y_1^*) + (1-\alpha)\mathcal{P}_0(x, y_1^*) - (\alpha_2 \mathcal{P}_2(x, y) + (1-\alpha)\mathcal{P}_0(x, y)) > 0$$

436 After rearranging we have

$$\alpha_1 \mathcal{P}_1(x) \geq \alpha_2 \mathcal{P}_2(x) * (\mathcal{P}_2(y|x) - \mathcal{P}_2(y_1|x)) + (1-\alpha)\mathcal{P}_0(x)(\mathcal{P}_0(y|x) - \mathcal{P}_0(y_1^*|x))$$

This must hold true for any $y$ so we can replace the rhs terms by $\Delta_{i,x}^j(y^*)$, in other words we have a sufficient condition for the size of $\alpha_1$ as

$$\alpha_1 \geq \frac{\mathcal{P}_2(x)}{\mathcal{P}_1(x)}\alpha_2\Delta_{1,x}^2(y_1^*) + \frac{\mathcal{P}_0(x)}{\mathcal{P}_1(x)}(1-\alpha)\Delta_{1,x}^0(y_1^*)$$

$$
\begin{aligned}
S(\alpha) &= \Pr_{x \sim \mathcal{P}_1} \{f(x) = y_1^*\} \\
&\geq \Pr_{x \sim \mathcal{P}_1} \left\{ \alpha_1 \geq \frac{\mathcal{P}_2(x)}{\mathcal{P}_1(x)}\alpha_2\Delta_{1,x}^2(y_1^*) + \frac{\mathcal{P}_0(x)}{\mathcal{P}_1(x)}(1-\alpha)\Delta_{1,x}^0(y_1^*) \right\} \\
&= \mathbb{E}_{x \sim \mathcal{P}_1} \mathbf{1}\left\{ \alpha_1 \geq \frac{\mathcal{P}_2(x)}{\mathcal{P}_1(x)}\alpha_2\Delta_{1,x}^2(y_1^*) + \frac{\mathcal{P}_0(x)}{\mathcal{P}_1(x)}(1-\alpha)\Delta_{1,x}^0(y_1^*) \right\} \\
&= \mathbb{E}_{x \sim \mathcal{P}_1} \mathbf{1}\left\{ 1 - \frac{\mathcal{P}_2(x)}{\mathcal{P}_1(x)}\frac{\alpha_2}{\alpha_1}\Delta_{1,x}^2(y_1^*) - \frac{\mathcal{P}_0(x)}{\mathcal{P}_1(x)}\frac{1-\alpha}{\alpha_1}\Delta_{1,x}^0(y_1^*) \geq 0 \right\} \\
&\geq \mathbb{E}_{x \sim \mathcal{P}_1} \left[ 1 - \frac{\mathcal{P}_2(x)}{\mathcal{P}_1(x)}\frac{\alpha_2}{\alpha_1}\Delta_{1,x}^2(y_1^*) - \frac{\mathcal{P}_0(x)}{\mathcal{P}_1(x)}\frac{1-\alpha}{\alpha_1}\Delta_{1,x}^0(y_1^*) \right] \\
&1 - \frac{\alpha_2}{\alpha_1}\mathbb{E}_{x \sim \mathcal{P}_1}\left[ \frac{\mathcal{P}_2(x)}{\mathcal{P}_1(x)}\Delta_{1,x}^2(y_1^*) \right] - \frac{1-\alpha}{\alpha_1}\mathbb{E}_{x \sim \mathcal{P}_1}\left[ \frac{\mathcal{P}_0(x)}{\mathcal{P}_1(x)}\Delta_{1,x}^0(y_1^*) \right] \\
&\geq 1 - \frac{\alpha_2}{\alpha_1}\mathcal{P}_2(\mathcal{X}_1)\Delta_1^2(y_1^*) - \frac{1-\alpha}{\alpha_1}\mathcal{P}_0(\mathcal{X}_1)\Delta_1^0(y_1^*)
\end{aligned}
$$

With the final line using the fact that the delta we max over all $x$'s

Now for distribution where $TV(P,P') \leq \epsilon_1$. By Lemma 11 in [HMMDZ23] we have that $\mathcal{P}'(y^*|x) > \mathcal{P}'(y|x)$ when $\mathcal{P}(y^*|x) > \mathcal{P}(y|x) + \frac{\epsilon_1}{1-\epsilon_1}$

We than write this as

$$\mathcal{P}(y_1^*|x)\mathcal{P}(x) > \mathcal{P}(y|x)\mathcal{P}(x) + \frac{\epsilon_1}{1-\epsilon_1}\mathcal{P}(x)$$

Following the same steps, procedure as before we get

$$\alpha_1 > \alpha_2\frac{\mathcal{P}_2(x)}{\mathcal{P}_1(x)}\left( \frac{(1-\epsilon_1)\Delta_2^1(y_1^*) + \epsilon_1}{1+2\epsilon_1} \right) + (1-\alpha)\frac{\mathcal{P}_0(x)}{\mathcal{P}_1(x)}\left( \frac{(1-\epsilon_1)\Delta_0^1(y_1^*) + \epsilon_1}{1+2\epsilon_1} \right)$$

Computing the $S(\alpha_1)$ again we get the condition we get

$$S(\alpha_1) \geq 1 - \frac{\alpha_2}{\alpha_1}\mathcal{P}_2(\mathcal{X}_1) \cdot \frac{(1-\epsilon_1)\Delta_2^1(y_1^*) + \epsilon_1}{1-2\epsilon_1} - \frac{1-\alpha}{\alpha_1}\mathcal{P}_0(\mathcal{X}_1)\frac{(1-\epsilon_1)\Delta_0^1(y_1^*) + \epsilon_1}{1-2\epsilon_1}$$

$\square$

# D   Proof of Theorem 4

We restate the theorem:

**Theorem** (Feature-only with two distributions). *Consider distribution $\mathcal{P}_1$ and $\mathcal{P}_2$ which are distributed by $h_1(x)$ and $h_2(x)$ respectively, where $x \sim \mathcal{P}_0$. Suppose there exist a $p$ such that*

$\mathcal{P}_0(y^*|x) \geq p, \forall x \in \mathcal{X}$. *Then success for the first collective against an $\epsilon_1$ classifier (against $P_1^*$) is lower bounded by*

$$S(\alpha_1) \geq 1 - \frac{\alpha_2}{\alpha_1} \cdot \frac{\mathcal{P}_2(X_1^*) \cdot \Delta_1^2(y^*)(1-\epsilon_1)}{p(1-\epsilon_1) - \epsilon_1} - \frac{1-\alpha}{\alpha} \mathcal{P}_0(\mathcal{X}_1) \cdot \frac{(1-p)(1-\epsilon_1) + \epsilon_1}{p(1-\epsilon_1) - \epsilon_1}$$

*Proof.* Similar to Theorem 3, we start with the $\epsilon = 0$. We require that, for any $x^* \in \mathcal{X}_1$ that $\mathcal{P}(y_1^*|x^*) > \mathcal{P}(y|x^*) \; \forall y \neq y_1^*$. This is equivalent to $\mathcal{P}(x^*, y_1^*) > \mathcal{P}(x^*, y)$ Our goal is find a sufficient condition for $\alpha_1$

We get a lower bound for $\mathcal{P}(x^*, y^*)$ by writing the mixture distribution as.

$$\mathcal{P}(x^*, y^*) = \alpha_1 \mathcal{P}_1(x^*, y^*) + \alpha_2 \mathcal{P}_2(x^*, y^*) + (1-\alpha)\mathcal{P}_0(x^*, y^*) \geq \alpha \mathcal{P}_1(x^*, y^*) + \alpha_2 \mathcal{P}_2(x^*, y^*)$$

where $\alpha = \alpha_1 + \alpha_2$

We also have when $y \neq y_1^*$ that

$$\mathcal{P}(x^*, y) = \alpha_1 \mathcal{P}_1(x^*, y) + \alpha_2 \mathcal{P}_2(x^*, y) + (1-\alpha)\mathcal{P}_0(x^*, y) = \alpha_2 \mathcal{P}_2(x^*, y) + (1-\alpha)\mathcal{P}_0(x^*, y)$$

Hence if
$$\alpha_1 \mathcal{P}_1(x^*, y^*) + \alpha_2 \mathcal{P}_2(x^*, y^*) \geq \alpha_2 \mathcal{P}_2(x^*, y) + (1-\alpha)\mathcal{P}_0(x^*, y)$$

than $\mathcal{P}(x^*, y^*) > \mathcal{P}(x^*, y)$

Rearranging we have

$$\alpha_1 \mathcal{P}_1(x^*, y^*) \geq \alpha_2 \mathcal{P}_2(x^*, y) - \alpha_2 \mathcal{P}_2(x^*, y^*) + (1-\alpha)\mathcal{P}_0(x^*, y)$$

$$\alpha_1 \mathcal{P}_1(x^*, y^*) \geq \alpha_2 \mathcal{P}_2(x^*)(\mathcal{P}_2(y|x^*) - \mathcal{P}_2(y^*|x^*)) + (1-\alpha)\mathcal{P}_0(y|x^*)\mathcal{P}_0(x^*)$$

$$\alpha_1 \geq \frac{\alpha_2 \mathcal{P}_2(x^*)(\mathcal{P}_2(y|x^*) - \mathcal{P}_2(y^*|x^*))}{\mathcal{P}_1(x^*, y^*)} + (1-\alpha)\frac{\mathcal{P}_0(y|x^*)\mathcal{P}_0(x^*)}{\mathcal{P}_1(x^*, y^*)}$$

For our bound on $\alpha_1$ we seek to maximize the rhs. We can do this by upper bounding $\mathcal{P}_2(y|x^*) - \mathcal{P}_2(y^*|x^*))$ by the suboptimality gap $\Delta_1^2$. By assumption we have that $\mathcal{P}_0(y^*|x) \geq p$ for all $x \in \mathcal{X}$ and hence $\mathcal{P}_0(y|x^*) \leq 1 - p$. For the denominator we note that $\mathcal{P}_1(x^*, y*) \geq \mathcal{P}_0(g^{-1}(x^*), y^*) \geq p\mathcal{P}_0(g^{-1}(x^*)$ Using this lowerbound in the denominator we get our required alpha must be at least

$$\alpha_1 \geq \alpha_2 \cdot \frac{\mathcal{P}_2(x^*)\Delta_1^2}{p\mathcal{P}_0(g^{-1}(x^*))} + (1-\alpha)\frac{(1-p)\mathcal{P}_0(x^*)}{p\mathcal{P}_0(g^{-1}(x^*))}$$

To compute the success rate we have

$$S(\alpha) = Pr_{x \sim \mathcal{P}_0^*}\{f(g(x)) = y^*\}$$

$$S = \sum_{x^* \in \mathcal{X}_1} \Pr_{x \sim \mathcal{P}_0^*}\{f(g(x)) = y^*|x \in g^{-1}(x^*)\} Pr_{x \in \mathcal{P}_0}\{x \in g^{-1}(x^*)\}$$

$$S = \sum_{x^* \in \mathcal{X}_1} \mathbf{1}\{f(x^*) = y^*\}\mathcal{P}_0(g^{-1}(x^*))\}$$

For a given fixed $x^*$ we have

$$\mathbf{1}\left\{f(x^*) = y^*\right\} \geq \mathbf{1}\left\{\alpha_1 \geq \alpha_2 \cdot \frac{\mathcal{P}_2(x^*)\Delta_1^2}{p\mathcal{P}_0(g^{-1}(x^*))} + (1-\alpha)\frac{(1-p)\mathcal{P}_0(x^*)}{p\mathcal{P}_0(g^{-1}(x^*))}\right\}$$

$$= \mathbf{1}\left\{1 - \frac{\alpha_2}{\alpha_1} \cdot \frac{\mathcal{P}_2(x^*)\Delta_1^2}{p\mathcal{P}_0(g^{-1}(x^*))} - \frac{(1-\alpha)}{\alpha_1}\frac{(1-p)\mathcal{P}_0(x^*)}{p\mathcal{P}_0(g^{-1}(x^*))} > 0\right\}$$

$$\geq 1 - \frac{\alpha_2}{\alpha_1} \cdot \frac{\mathcal{P}_2(x^*)\Delta_1^2}{p\mathcal{P}_0(g^{-1}(x^*))} - \frac{(1-\alpha)}{\alpha_1} \frac{(1-p)\mathcal{P}_0(x^*)}{p\mathcal{P}_0(g^{-1}(x^*))}$$

Computing the summation we get

$$Pr_{x \sim \mathcal{P}_0^*}\{f(g(x)) = y^*\}$$

$$= 1 - \sum_{x^* \in \mathcal{X}_1} \frac{\alpha_2}{\alpha_1} \cdot \frac{\mathcal{P}_2(x^*)\Delta_1^2}{p\mathcal{P}_0(g_1^{-1}(x^*))} \cdot \mathcal{P}_0(g_1^{-1}(x^*)) - \sum_{x^* \in \mathcal{X}_1} \frac{(1-\alpha)}{\alpha_1} \frac{(1-p)\mathcal{P}_0(x^*)}{p\mathcal{P}_0(g^{-1}(x^*))} \cdot \mathcal{P}_0(g_1^{-1}(x^*))$$

$$\geq 1 - \frac{\alpha_2}{p\alpha_1}\mathcal{P}_2(\mathcal{X}_1)\Delta_1^2 - \frac{(1-\alpha)\cdot(1-p)}{p\alpha_1}\mathcal{P}_0(X^*)$$

Which gives us the $\epsilon = 0$ case.

When considering $\epsilon > 0$, we again cite Lemma 11 in [HMMDZ23], that: $\mathcal{P}(x^*, y^*) > \mathcal{P}(x^*, y) + \frac{\epsilon}{1-\epsilon}\mathcal{P}(x^*)$

Expanding all terms out with the mixture distribution we have

$$\mathcal{P}(x^*, y) = \alpha\mathcal{P}_1(x^*, y^*) + \alpha_2\mathcal{P}_2(x^*, y^*) + (1-\alpha_1-\alpha_2)\mathcal{P}_0(x^*, y^*) \geq \alpha\mathcal{P}_1(x^*, y^*) + \alpha_2\mathcal{P}_2(x^*, y^*)$$

Plugging this into our expression we have

$$\alpha\mathcal{P}_1(x^*, y^*) + \alpha_2\mathcal{P}_2(x^*, y^*) > \alpha_2 * \mathcal{P}_2(x^*, y) + (1-\alpha)\mathcal{P}_0(x^*, y) + \frac{\epsilon}{1-\epsilon}(\alpha\mathcal{P}_1(x^*) + \alpha_2\mathcal{P}_2(x^*) + (1-\alpha)\mathcal{P}_0(x^*))$$

Using conditionals we can rewrite this as

$$\alpha_1\mathcal{P}_1(y^*|x^*)\mathcal{P}_1(x^*) + \alpha_2\mathcal{P}_2(y^*|x^*)\mathcal{P}_2(x^*) >$$
$$\alpha_2 * \mathcal{P}_2(y|x^*)\mathcal{P}_2(x^*) + (1-\alpha)\mathcal{P}_0(y|x^*)\mathcal{P}_0(x^*) + \frac{\epsilon}{1-\epsilon}(\alpha_1\mathcal{P}_1(x^*) + \alpha_2\mathcal{P}_2(x^*) + (1-\alpha)\mathcal{P}_0(x^*))$$

Rearranging with $\alpha_1$ we get

$$\alpha_1 > \alpha_2\frac{\mathcal{P}_2(x^*)}{\mathcal{P}_1(x^*)} \cdot \frac{(\mathcal{P}_2(y|x^*) - \mathcal{P}_2(y^*|x^*))}{\mathcal{P}_1(y^*|x^*) - \frac{\epsilon}{1-\epsilon}} + (1-\alpha)\frac{\mathcal{P}_0(x^*)}{\mathcal{P}_1(x^*)} \cdot \frac{(\mathcal{P}_0(y|x^*) + \frac{\epsilon}{1-\epsilon})}{(\mathcal{P}_1(y^*|x^*) - \frac{\epsilon}{1-\epsilon})}$$

We note that $\mathcal{P}_2(y|x^*) - \mathcal{P}_2(y^*|x^*)$ can be upper bounded by $\Delta_1^2(y^*)$ We can also upper bound and lower bound the numerators and denominators with the following expressions

$\mathcal{P}_1(x^*) \geq \mathcal{P}_0(g^{-1}(x^*))$ as every point $x^*$ for $\mathcal{P}_1$ is mapped from some point in $\mathcal{P}_0$ and likewise $\mathcal{P}_1(y^*|x^*) \geq \mathcal{P}_0(y^*|g_1^{-1}(x^*)) \geq p$ From the support of $P_0(y_1^*|x) \geq p$ we have $\mathcal{P}_0(y|x^*) \leq 1 - p$ for $y \neq y_1^*$

All together we get

$$\alpha_1 > \alpha_2\frac{\mathcal{P}_2(x^*)}{\mathcal{P}_0(g_1^{-1}(x^*))} \cdot \frac{\Delta_1^2(y^*)}{p - \frac{\epsilon}{1-\epsilon}} + (1-\alpha)\frac{\mathcal{P}_0(x^*)}{\mathcal{P}_0(g_1^{-1}(x^*))} \cdot \frac{(1-p+\frac{\epsilon}{1-\epsilon})}{(p - \frac{\epsilon}{1-\epsilon})}$$

With this we can compute the expected value as above

For a given fixed $x^*$ we have

$$\mathbf{1}\left\{f(x^*) = y^*\right\} \geq \mathbf{1}\left\{\alpha_1 > \alpha_2\frac{\mathcal{P}_2(x^*)}{\mathcal{P}_0(g_1^{-1}(x^*))} \cdot \frac{\Delta_1^2(y^*)}{p - \frac{\epsilon}{1-\epsilon}} + (1-\alpha)\frac{\mathcal{P}_0(x^*)}{\mathcal{P}_0(g_1^{-1}(x^*))} \cdot \frac{(1-p+\frac{\epsilon}{1-\epsilon})}{(p - \frac{\epsilon}{1-\epsilon})}\right\}$$

$$= \mathbf{1}\left\{1 - \frac{\alpha_2}{\alpha_1}\frac{\mathcal{P}_2(x^*)}{\mathcal{P}_0(g_1^{-1}(x^*))}\cdot\frac{\Delta_1^2(y^*)}{p - \frac{\epsilon}{1-\epsilon}} - \frac{1-\alpha}{\alpha_1}\frac{\mathcal{P}_0(x^*)}{\mathcal{P}_0(g_1^{-1}(x^*))}\cdot\frac{(1 - p + \frac{\epsilon}{1-\epsilon})}{(p - \frac{\epsilon}{1-\epsilon})} > 0\right\}$$

$$\geq 1 - \frac{\alpha_2}{\alpha_1}\frac{\mathcal{P}_2(x^*)}{\mathcal{P}_0(g_1^{-1}(x^*))}\cdot\frac{\Delta_1^2(y^*)}{p - \frac{\epsilon}{1-\epsilon}} - \frac{1-\alpha}{\alpha_1}\frac{\mathcal{P}_0(x^*)}{\mathcal{P}_0(g_1^{-1}(x^*))}\cdot\frac{(1 - p + \frac{\epsilon}{1-\epsilon})}{(p - \frac{\epsilon}{1-\epsilon})}$$

Plugging back in we get

$$Pr_{x\sim\mathcal{P}_0^*}\{f(g(x)) = y^*\}$$

$$= 1 - \sum_{x^*\in\mathcal{X}_1}\frac{\alpha_2}{\alpha_1}\frac{\mathcal{P}_2(x^*)}{\mathcal{P}_0(g_1^{-1}(x^*))}\cdot\frac{\Delta_1^2(y^*)}{p - \frac{\epsilon}{1-\epsilon}}\cdot\mathcal{P}_0(g_1^{-1}(x^*)) - \sum_{x^*\in\mathcal{X}_1}\frac{1-\alpha}{\alpha_1}\frac{\mathcal{P}_0(x^*)}{\mathcal{P}_0(g_1^{-1}(x^*))}\cdot\frac{(1 - p + \frac{\epsilon}{1-\epsilon})}{(p - \frac{\epsilon}{1-\epsilon})}\cdot\mathcal{P}_0(g_1^{-1}(x^*))$$

$$\geq 1 - \frac{\alpha_2}{\alpha_1}\cdot\frac{\mathcal{P}_2(X^*)\cdot\Delta_1^2(y^*)}{p - \frac{\epsilon}{1-\epsilon}} - \frac{1-\alpha}{\alpha}\mathcal{P}_0(X^*)\cdot\frac{1 - p + \frac{\epsilon}{1-\epsilon}}{p - \frac{\epsilon}{1-\epsilon}}$$

$$\geq 1 - \frac{\alpha_2}{\alpha_1}\cdot\frac{\mathcal{P}_2(X^*)\cdot\Delta_1^2(y^*)(1-\epsilon)}{p(1-\epsilon) - \epsilon} - \frac{1-\alpha}{\alpha}\mathcal{P}_0(X^*)\cdot\frac{(1 - p)(1-\epsilon) + \epsilon}{p(1-\epsilon) - \epsilon}$$

$\square$

## E   Extension to many collectives

We can extend the mixture distribution to include an arbitrary number of distributions. This can be written as

$$\mathcal{P} = \sum_{i=1}^n \alpha_i\mathcal{P}_i + (1 - \alpha)\mathcal{P}_0$$

where $\alpha = \sum_{i=1}^n \alpha_i$

By following the same procedure in Appendix C we provide a claim for success for collective one against an arbitrary number of distributions.

**Theorem 5.** *The success of collective action for the feature-label strategy against many distributions is lower bounded by*

$$S(\alpha_1) \geq 1 - \frac{1-\alpha}{\alpha_1}\mathcal{P}_0(\mathcal{X}_1)\frac{(1 - \epsilon_1)\Delta_0^1(y_1^*) + \epsilon_1}{1 - 2\epsilon_1} - \sum_{i=2}^n \frac{\alpha_i}{\alpha_1}\mathcal{P}_i(\mathcal{X}_1)\cdot\frac{(1 - \epsilon_1)\Delta_i^1(y_1^*) + \epsilon_1}{1 - 2\epsilon_1}$$

*Proof.* We repeat the same expansion in Appendix C for an arbitrary number of distributions.

$$\mathcal{P} = \sum_{i=1}^n \alpha_i\mathcal{P}_i + (1 - \alpha)\mathcal{P}_0$$

The rest follows the same procedure, as each additional mixture term can be handled independently.

$\square$

We can do the same for feature-only strategies.

**Theorem 6.** *The success of collective action for the feature-only strategy against many distributions is lower bounded by*

$$\geq 1 - \frac{1-\alpha}{\alpha}\mathcal{P}_0(X^*)\cdot\frac{(1 - p)(1-\epsilon) + \epsilon}{p(1-\epsilon) - \epsilon} - \sum_{i=2}^n \frac{\alpha_i}{\alpha_1}\cdot\frac{\mathcal{P}_i(\mathcal{X}_1)\cdot\Delta_1^i(y_1^*)(1-\epsilon)}{p(1-\epsilon) - \epsilon}$$

*Proof.* We repeat the same expansion in Appendix D for an arbitrary number of distributions with the same expansion of the mixture distribution and the rest follows the same procedure, as each additional mixture term can be handled independently.

$\square$

