# OpenReview forum: "Sync or Sink: Bounds on Algorithmic Collective Action with Noise and Multiple Groups"
_NeurIPS.cc/2025/Conference — Submitted to NeurIPS 2025_

### Official Review · Reviewer_yFBj · 2025-06-02

**Clarity:** 3
**Significance:** 2
**Originality:** 2
**Rating:** 4
**Confidence:** 4

**Summary:**

In collective action, a group of users strategically modifies their own data to inject a correlation between the transformation and a target label in learned classifier. This paper builds on top of previous works and provides a theoretical framework for a multi-collective setting, by combining the theoretical framework for a single collective with empirical results for multiple collectives.
The paper focuses on theoretical lower bounds for the success of a collective in a two-collectives setting, but provides a generalized bound for n collective as well.
They then use this framework to model noisy strategies, simulating users who fail to follow the coordinated strategy perfectly, and then empirically test the effect of different types of noises on the success.

**Questions:**

1. As mentioned above, the motivation for label noise is unclear. What is the rationale for introducing label noise?
2. In the experiment with the the random characters subset, why were these specific characters chosen? Is the choice of characters meaningful, or only the size of the subset? Did you investigate how varying the subset size affect the success?
3. What exactly is randomized in each experiment run? Are the collectives randomized per run?
4. The standard deviations in the figures are quite large. Is there an intuitive explanation for this variability? What are the causes for the variability?
5. In section 7.1, you mention an example where two collectives with different $g$ and $ y^* $ that still hinder each other's success. This is counterintuitive. What is the context and how do the collectives interfere each other? You explain that it is because of *character modifications that look sufficiently “close”*. What does *"close"* mean in this context and how does it lead to interference?
6. This paper models the population and noise sources as distributions. I'm not sure if that's the best modeling choice. Would a finite-sample approach, such as in [1] be more suitable to discuss miscommunication?

[1] Gauthier, Etienne, Francis Bach, and Michael I. Jordan. "Statistical Collusion by Collectives on Learning Platforms." arXiv preprint arXiv:2502.04879 (2025).

**Ethical Concerns:**

["NO or VERY MINOR ethics concerns only"]

**Final Justification:**

After discussing with the authors, I am more convinced of the value of the paper and it academical importance.
However, I increased my score to 4 and not 5 because I do not think the impact and contribution are enough, and because, as the other reviewers noticed, the connection to ML is made clear only in section 3, and until then relies on being familiar with prior work.

The theoretical contribution is a straightforward generalization of prior work, the experimental results only confirm what one could expect, and I think the suggested future work will not inspire much further research without concrete real world case studies which are not discussed.

**Limitations:**

yes

**Paper Formatting Concerns:**

Several paragraphs omit the line numbering (e.g. 113-114, 138-139...), but I did not notice major formatting concerns.

**Quality:**

4

**Strengths And Weaknesses:**

Strengths:
1. The paper provides a generalization of the lower bound on a collective success in a multi-collective settings, which natural and valuable extension of prior work.
2. The writing is clear, and the structure of the paper, including the background and problem formulation, makes the subject accessible.
3. Figures are well designed and effectively convey the message of the research questions.

Weaknesses:
1. While the generalized success bound is important to formalize, its derivation is very similar to the single collective analysis and does not yield surprising insight or fundamental understanding.
2. The experimental settings are limited. The paper reuses the same setup from prior work without exploring new scenarios that may be more fitting for a multi-collective setting. For example, the instruction "Place character 'A' every 20 words" is simple and unlikely to be misunderstood. More ambiguous or realistic problems would be more appropriate to test the theory.
3. The concept of the label noise experiments is unclear. If the goal is to create a correlation between a signal $g$ and a target label $ y^* $, then it is unclear why $ y^* $ would be noisy. If the members of the collective misunderstood the the goal itself, it is unlikely they would have joined the collective in the first place. It would help if the authors justified where label noise arises, or find more justified settings, especially since the key experimental results (Fig. 3A) depend on it. In the current context, a more suiting interpretation is a multi-collective problem where every collective has a similar $g$ but different $y^*$, which is a different problem with a different meaning of the result.
4. Several details of the experiment design are not fully explained (see questions section below).

I would like to accept the paper, as it is valuable to have this generalization formally documented. However, in the current form I believe the paper does not contribute enough beyond prior work to warrant acceptance. The theory is sound, but relatively straightforward. To help me raise the score to accept, I think the paper should offer new experimental settings (especially ones which are more plausible for inter-collective miscommunication), and a clearer justification for the noise sources, particularly label noise.


Presentation comments (do not affect my scoring):
1. Figure fonts are very small. Please enlarge for readability.
2. Several paragraphs omit the line numbering (e.g. 113-114, 138-139...). You should look into that.
3. Inline math is often followed by a missing punctuation or spacing (e.g. lines 115, 198...).
4. Line 211 refers to figure axes before naming the actual figure. Rephrasing will improve quality.

---

> ### Author Rebuttal · Authors · 2025-07-30
>
> We thank the reviewer for their deep engagement with the work. The feedback has made the paper stronger and we will swiftly incorporate improvements to the typesetting, including figure size, line numbers, figure references etc. We will start by addressing some of the high level concerns, combining where appropriate, and then address specific concerns.
>
> _The concept of the label noise experiments is unclear. If the goal is to create a correlation between a signal $g$ and a target label $ y^* $, then it is unclear why $ y^* $ would be noisy. If the members of the collective misunderstood the the goal itself, it is unlikely they would have joined the collective in the first place. It would help if the authors justified where label noise arises, or find more justified settings, especially since the key experimental results (Fig. 3A) depend on it._
>
>
> *As mentioned above, the motivation for label noise is unclear. What is the rationale for introducing label noise?*
>
> [Q1, W3] We see why the rationale for the noise label experiments can be confusing. In terms of ecological justification, the stated goal of a collective can be more vague that the specific implementation - for example, a group that’s dedicated to supporting an artist, may actually have many specific songs, etc that they could choose from and there can be coordination gaps from picking the correct song, or members of the group erroneously thinking that doing the same changes for all songs would be beneficial, which may or may not be the case. Being aligned at a high level ‘goal’ may not directly translate to being coordinated on the specific action.
>
> Another justification would be more malicious actors, who want to thwart the collective, by joining the group to learn the 'signal’ and apply that to other labels. As the reviewer also alludes to, it can also be used to help make the connection between the noise views of the system vs the multiple collectives, where that distinction is a bit more concrete. The noise on the label also is a term that comes out of the bounds we proved which is different from prior bounds, so we believe including them is justified.
>
> *While the generalized success bound is important to formalize, its derivation is very similar to the single collective analysis and does not yield surprising insight or fundamental understanding.*
>
> [W1] We agree that the derivation is very similar to the single collective analysis, the key contribution is formalizing this notion of overlap between, not just the baseline distribution and the single collective’s distribution, but across multiple distributional sources. We believe formalizing this is merited.
>
> *The experimental settings are limited. The paper reuses the same setup from prior work without exploring new scenarios that may be more fitting for a multi-collective setting. For example, the instruction "Place character 'A' every 20 words" is simple and unlikely to be misunderstood. More ambiguous or realistic problems would be more appropriate to test the theory.*
>
> [W2] In terms of choice of experiment, we choose to limit them to more well documented cases to have a more direct comparison to both [HMMDZ23, KVKS25], which provide bounds [HMMDZ23] or examined multiple collectives, which can be explored in the bounds we provide. While more complex systems can introduce more ambiguity, even “simple" instructions, when scaled to the level of online communities, can have coordination challenges.. [XZF+25] discusses some of the challenges that organizers face and inherent issues in collective coordination.
>
> *In the experiment with the random characters subset, why were these specific characters chosen? Is the choice of characters meaningful, or only the size of the subset? Did you investigate how varying the subset size affects the success?*
>
> [Q2] The specific initial characters came from prior work [KVKS25], and then we expanded to a small subset by considering other characters that did not appear in the dataset (which was also a criteria shown in prior work). [HMMDZ23] in their paper mention mostly that the individual characters were relatively unimportant so we mostly looked at randomized within a small subset vs large (aka size). To show a more smooth change, we will add experiments that vary the size of the set more gradually.
>
>
> *What exactly is randomized in each experiment run? Are the collectives randomized per run?*
> *The standard deviations in the figures are quite large. Is there an intuitive explanation for this variability? What are the causes for the variability?*
>
> [Q3, Q4] For randomization, the collective members are randomized each round, this can lead to some of the higher variability/standard deviations found in the figures.
>
> *In section 7.1, you mention an example where two collectives with different and that still hinder each other's success. This is counterintuitive. What is the context and how do the collectives interfere each other? You explain that it is because of character modifications that look sufficiently “close”. What does "close" mean in this context and how does it lead to interference?*
>
> [Q5] In 7.1 we refer to [KVKS25] Figure 3d. When looking at the specific characters chosen in that setup with the specific tokenizer (DistilBERT), the embedding of a resume with character 1 modification and character 2 modification end up being nearly identical. which can lead to interference. We will add those specific details in the paper.
>
> *This paper models the population and noise sources as distributions. I'm not sure if that's the best modeling choice. Would a finite-sample approach, such as in [1] be more suitable to discuss miscommunication?*
>
> [Q6] Other modeling choices, such as finite sampling methods could be an interesting view. [1], however, primarily focuses on how a collective estimates its effectiveness over timestamps.. There, the sequential aspect is key. This is a slightly different, yet complementary, setup than the one we’re considering. Our focus on distributions was to help characterize a ‘final state’ outcome across multiple distinct distributions. We can add some of the more specific possible modeling choices within the paper to highlight some of the similarities and differences between them.

---

> > ### Comment · Reviewer_yFBj · 2025-08-01
> >
> > I thank the authors for addressing my concerns. I find the clarification of the label noise academically satisfactory and decided to increase my score. However, I think it will strengthen the paper to provide evidence (as case studies or news coverage) for unsynchronized collectives in the real world. I acknowledge it is a difficult task since we may be aware of successful collective actions, but not failed ones.
> >
> > I would like to visit the example of malicious actors who want to thwart the collective. If the malicious actors are also acting collectively, these actors could implement the "erasing a signal" strategy from [HMMDZ23] to erase the signal that the original collective is attempting to plant. Depending on the problem setting, there may be a better strategy for the malicious actors than using random labels. Studying the best strategy for the malicious actors diverges from the current work, but the point is that malicious actors could serve as an example for label noise only if they are uncoordinated (unless proved that their best strategy is to randomize labels).
> >
> > Regarding the experiments, I thank the authors for directing to the relevant sources. However, in my personal opinion a paper should be mostly self-contained and explicitly mention the experiment details and reasonings.
> > I also appreciate the authors' wish to rely on established experiments from prior work. But seeing how collective action is an emerging field, I believe it will be helpful to the community to suggest and establish more experimental settings.
> >
> > Lastly, your explanation of using distributions rather than the finite modeling is convincing. Adding the modeling discussion to the paper could be interesting, but is not needed.

---

> > > ### Author Response · Authors · 2025-08-07
> > >
> > > We thank the reviewer for continued engagement with the work!
> > >
> > > *Real world unsynchronized collectives*
> > >
> > > We agree both that it would be good to have and a bit more difficult since there’s fewer documented examples of it: this is explicitly called out as a challenge in [XZF+25]  about how there is far less discussion about failed examples. There are a few somewhat tangentially related examples we can discuss.There are more cases about things like astroturfing failing to make grounds in certain instances [2] but the measure of success is a bit less defined. However, they do claim that less coordinated groups had less impact even than more tightly coordinated ones. One of the most comparable examples include google-bombing [3] (where a group tried to associated a phrase with George W Bush which was initially successful but lost momentum in the face of algorithmic changes), or challenges in encouraging in advocating for privacy
> > >
> > > [2]  Tardelli, S., Nizzoli, L., Avvenuti, M. et al. Multifaceted online coordinated behavior in the 2020 US presidential election. EPJ Data Sci. 13, 33 (2024). https://doi.org/10.1140/epjds/s13688-024-00467-0
> > >
> > > [3] Baruchson‐Arbib, Shifra, and Judit Bar‐Ilan. "Manipulating search engine algorithms: the case of Google." Journal of Information, Communication and Ethics in Society 5, no. 2-3 (2007): 155-166.
> > >
> > > *Malicious actors using an erasing the signal strategy*
> > >
> > > We agree that looking at how “erasing a signal” from [HMMDZ23] could be an important analysis in context for how a malicious (or just opposing) collective may interfere with another group. As our bounds presented don’t actually specify the behavior of the 2nd collective, we can leverage much of the same work to derive the interaction effect of having one employing a signal planting strategy and having the 2nd employing an erasing strategy can affect the success rate.
> > >
> > > We will sketch out here how we can analyze this case using our existing bounds/extensions to the signal erasing case. For brevity, we will consider the $\epsilon_1 = 0$ case though the extension follows in a similar manner. We can add a version of this in the Appendix.
> > >
> > > Let collective one be planting a signal with g and collective two be trying to erase that same signal.
> > > We showed our bound for two collectives as
> > >
> > > $S(\alpha_1) \geq 1 - \frac{\alpha_2}{\alpha_1} \mathcal{P}_2(\mathcal{X}_1)\Delta_1^2(y_1) - \frac{1-\alpha}{\alpha_1} \mathcal{P}_0(\mathcal{X}_1) \Delta_1^0(y_1)$
> > >
> > > (Because of typeseting challenges in openreview, $y_1 = y_1^*$ from the paper)
> > >
> > > Notably we didn’t make any explicit assumptions about the form of the 2nd collective. We can then consider the case where the second collective is trying to erase this signal.
> > > In particular we can focus on the
> > > $\Delta_1^2(y_1)$
> > > which is defined as
> > > $\Delta_1^2(y_1) = \max_{x \in X_1} (\max_{y \in \mathcal{Y}} \mathcal{P}_2(y | x) - \mathcal{P}_2 (y_1 | x))$
> > >
> > >
> > > This term represents the max difference in classification between the distribution points in $X_1$ and the distribution induced by collective $2$. For the signal erasing strategy, we note that the that collective plants points of the form $h_2(z) = (x, argmax_{y \in \mathcal{Y}} P_0(y | g(x)))$, In other words, it takes points in $x$ and sets it’s own y to be what $g(x)$ would be classified by in the baseline distribution ($P_0$).
> > >
> > > In this way, unless all points $g(x)$ in the original distribution map to $y_1$ then $\Delta_{1}^2(y_1) = 1$.
> > > Hence this represents a large impediment to collective one, where $\Delta_1^2(y_1)$ is at its max value at $1$ in this direct confrontation case.
> > >
> > > We can also look at how the presence of signal planting collectives affect the efficacy of signal erasing collectives. We note that this is not in our paper but can add this to the appendix (which would follow a similar derivation to the two collective case in the paper).
> > >
> > > Using that setup we can find that success of signal erasing (assuming collective two is doing the erasing) is $$S(\alpha_2) \geq 1 - \frac{\alpha_1}{\alpha_2} max_{y \in Y}  (P_1(y | x) - P_1(y_2(x) | x) ) + 2 \frac{ 1- \alpha}{\alpha} \tau$$
> > > Where $y_2^*(x) = argmax_{y \in \mathcal{Y}} P_0(y|g(x))$ and $\tau = \max_{y \in \mathcal{Y}} | P_0(y | x) - \mathcal{P}_0(y | g(x))|$, and $g(x)$ is the signal.
> > >
> > > To understand the impact of collective one’s signal planting activity, we examine the term
> > > $$\max_{y \in Y} ( P_1(y | x) - P_1(y_2(x) | x)) $$
> > > If there is at least one point where $y_2(x) \neq y_1$ then this term equals one and we get the reduced efficacy being. $1 - \frac{\alpha_1}{\alpha_2} - \frac{2 (1-\alpha)}{\alpha} \tau$, showing that this type of conflict between these groups using the same signal decreases the chance of success for both of them.
> > >
> > >
> > > *Clarification on experiment choices*
> > >
> > > We will provide more explicit details of the choice of experiments and rationale to be more self-contained. We can also add some suggestions of other experiments that would be important going forward!

---

> > > > ### Comment · Reviewer_yFBj · 2025-08-08
> > > >
> > > > I thank the authors for their thoughtful response and effort, and for the additional references. I believe that including the discussion about the real-world cases and the analysis of the opposing collective using the paper's framework (even if mostly in the appendix) will highlight the strength of the framework and make the paper stronger.
> > > >
> > > > Regarding additional experiments, they should be included only if they are are performed, not suggested. However, suggestions for case studies, if there are any, should be included.
> > > >
> > > > Unrelated, I noticed that the main concern of other reviewers is that the paper is not self-contained and does not introduce algorithmic collective action well. It may help to respond with the text the authors will add or change in the paper to better explain the subject.

---

> > > > > ### Author Response · Authors · 2025-08-08
> > > > >
> > > > > We thank the reviewer for their continued engagement.
> > > > >
> > > > > We will incorporate that discussion about the real-world cases in the main body and the opposing collective framing. We will mention the ability to use this analysis for opposing collectives towards the end Section 4 towards the end while adding the bulk of the analysis in the Appendix. Thank you for the suggestions!
> > > > >
> > > > > We will include suggestions for future case studies in the Discussion.
> > > > >
> > > > > Noted on the concerns of the other reviewers. We have responded with some of the suggested changes to the text and will add more. Thanks for the suggestions and engagement with the others!

---

### Official Review · Reviewer_BN7G · 2025-07-03

**Clarity:** 2
**Significance:** 3
**Originality:** 2
**Rating:** 4
**Confidence:** 2

**Summary:**

The paper contributes to the line of work on algorithmic collective action, by moving away from perfect coordination. In the algorithmic collective action problem a group of "manipulators", each holding a single data point, want to fool a classifier. The paper offers both theoretical upper bounds and experiments.

**Questions:**

Can you elaborate on the role of signal and strategy in your model?

**Ethical Concerns:**

["NO or VERY MINOR ethics concerns only"]

**Final Justification:**

The authors answered some of my questions about the model, and promised to revise the paper to justify their (and their predecessors') modeling choices.

**Limitations:**

addressed in the paper

**Paper Formatting Concerns:**

A few missing line numbers

**Quality:**

3

**Strengths And Weaknesses:**

I found the formal model and the presentation quite confusing. While I get the intuition that the authors are trying to capture, it is not clear to me why it is captured by the formal model.

First, the strategy being a mapping Z\to Z is a non-obvious choice. In particular, it presumes that the strategy is anonymous, i.e., all manipulators act in the same way given the same inputs. This is not without loss of generality, but probably appropriate for the settings studied in this work. Still, worth a comment.

More importantly, I do not understand the point of a signal and how it is different from a strategy. Given the motivating example, I expected the manipulator's goal to get the classifier to output a specific label on a certain subset of inputs, but this seems different? The definition of success is then in terms of the signal, not in terms of the strategy. What is the role of strategy then?

The paper provides some lower bounds on success rates; these are complicated expressions, so I am not sure how useful they are. Also, the paper does not say if they are tight. Also, the paper does not explicitly analyze the difference between these bounds and the bounds for the coordinated case from prior work. Overall, I found it very hard to evaluate the contribution, because the model is not well-explained; also the paper does not seem to be proofread properly.

A few minor issues:

71: success->success of

In the unnumbered chunk between 113 and 114, "Where" should not be capitalized

also, 3 lines below 113 there is a missing whitespace

128: something is missing here

130 one's->1's

133 is -> be

136 missing full stop

Display formula below 138: what is \mathcal X^*?

176: the explanation of what r is is confusing

183: how do you implement "arbitrary location"?

198: missing whitespace

Table 1, last line: remove one of "place" and "insert"

202-203: why are the ranges so different for the two experiments?

220-221: this sentence does not parse

Theorem 1: epsilon_1 -> epsilon_1-suboptimal

---

> ### Author Rebuttal · Authors · 2025-07-30
>
> We thank the reviewer for their comments. We have swiftly addressed all the drafting errors. We will add the reviewers' concerns below and address them inline.
>
>
> *I found the formal model and the presentation quite confusing. While I get the intuition that the authors are trying to capture, it is not clear to me why it is captured by the formal model.First, the strategy being a mapping Z\to Z is a non-obvious choice.*
>
>
>
> With regards to the clarity of the formalization, we took the initial setup from [HMMDZ23], and aimed to be faithful to the prior work. However, we recognize this setup can be confusing and we will sketch out how we plan to clarify the differences and improve the initial setup, which we will add to the main body of the paper.
>
> We have $\mathcal{X}$ as the input features, $\mathcal{Y}$ as the label, and $\mathcal{Z} = (\mathcal{X}, \mathcal{Y})$ - *i.e* just the feature, label tuple. A strategy, in short, maps the a tuple $z=(x, y) \in  \mathcal{Z}$ where $x \in \mathcal{X}$, and $y \in \mathcal{Y})$ to a new tuple $z’ = (x’, y’) \in \mathcal{Z}$
>
> The most noticeable part of the strategy is how feature $x \in \mathcal{X}$ will change, and hence we call this out as the “signal” operationalized by function $g$. In the feature-label strategy, therefore, the members of the collective implement strategy $h_1(x, y) = (g_1(x), y^*)$ by using signal function $g_1$ has the function that modifies the feature input $x$.  For the feature-only strategy the members of the collectives would only implement strategy $h_2(x, y) =  (g_2(x), y)$ where $g_2$ is the signal function modifying input feature $x$.
>
> Ultimately this is meant to capture the difference between feature-label strategies and feature-only strategy and more broadly, formalize a strategy as a combination of signal planting + potential feature changing.
>
> *More importantly, I do not understand the point of a signal and how it is different from a strategy. Given the motivating example, I expected the manipulator's goal to get the classifier to output a specific label on a certain subset of inputs, but this seems different? The definition of success is then in terms of the signal, not in terms of the strategy. What is the role of strategy then?*
>
> In addition to what’s discussed above, (namely a strategy may operate on the feature + label while the signal refers to the operations on just the feature), the distinction between the strategy and the signal can also be helpful to separate out the difference between training and inference. A collective implements an entire strategy at training time (which may or may not involve control and changing of labels), so that at inference time, anyone with the same ``signal” will get a specific outcome from the ML model. We will clarify this in the paper.
>
> *The paper provides some lower bounds on success rates; these are complicated expressions, so I am not sure how useful they are. Also, the paper does not say if they are tight. Also, the paper does not explicitly analyze the difference between these bounds and the bounds for the coordinated case from prior work.*
>
> We discuss the high level takeaways from these bounds in terms of what terms/relationships are important, deferring the details of the proof to the appendix so as to not detract from the main conceptual takeaways. We discuss the differences between prior work and our work, in lines 151-154. We can also expand upon the Figure 1 with an explicit $r = 0$, which would be the perfectly coordinated case.
>
> *In particular, it [the author's formalization] presumes that the strategy is anonymous, i.e., all manipulators act in the same way given the same inputs. This is not without loss of generality, but probably appropriate for the settings studied in this work. Still, worth a comment.*
>
> We acknowledge that there is an assumption that the collective attempts to act in the same way given the same input; this is similar to approaches taken in prior work. However, we do note our work, which,while primarily framed in terms of noise, can also be applied to when members of a collective are not implementing the same exact strategy, ie, we could consider one large collective as being any number of collectives implementing their individual strategies towards the same goal, including differing distributions. While this is not quite the full analysis, this work helps to bridge that gap.
>
> Some other clarifications are below:
>
> *Display formula below 138: what is \mathcal X^*?*
>
> That should be just $\mathcal{X_i}$. Apologies!
>
> *The explanation of what r is is confusing*
>
> We can simplify $r$ is just $\frac{\alpha_1}{\alpha}$ in most contexts here. For example,, if a collective size $\alpha = 0.01\$ and $20$% of the group is experiencing noise, then $\alpha_1 = 0.2$, $\alpha_2 = 0.8$ and $r = 0.2$ or $20$%
>
> *How do you implement "arbitrary location"?*
>
> We sample a number of times to insert a character from a uniform distribution U(0, K), and after choosing a number of insertions. After picking the number of times we need to insert a character, we look at all possible locations to place a new character (aka look at the number of spaces between words available in a given piece of text) and sample without replacement from there. Importantly, this type of manipulation is meant to be very generic (i.e., this type of insertion could be applied to many kinds of classification inputs).
>
> *Why are the ranges so different for the two experiments?*
>
> In [HMMDZ23] the effectiveness range of a feature-label strategy vs feature-only strategy is quite substantial - this is because controlling both the feature and a label gives a lot more control for the collective, and hence, easier for a small group to manipulate. We can add that the choice of the range was drawn from these results in [HMMDZ23] and reflect the inherent differences between the two of them.

---

> > ### Comment · Reviewer_BN7G · 2025-08-07
> >
> > Thanks for clarifying, this was helpful. It is still not immediate to me why the aim is to benefit (at inference time) those with a specific signal (rather than specific inputs). I think the paper would benefit from a concrete example illustrating all these points. I acknowledge that you inherited the quirks of the model from the predecessor paper, but presumably the predecessor paper motivated them clearly; to make the paper self-contained not just in a technical, but in a conceptual sense, you'd have to do the same.

---

> > > ### Author Response · Authors · 2025-08-07
> > >
> > > We thank the reviewer for their response!
> > >
> > > We will add concrete examples of this distinction in Section 3 to help clarify why distinguishing between them can be helpful. Thank you for the suggestion!

---

### Official Review · Reviewer_Kze9 · 2025-07-03

**Clarity:** 3
**Significance:** 2
**Originality:** 3
**Rating:** 3
**Confidence:** 2

**Summary:**

This paper provides guarantees on the success of collective action in the presence of both coordination noise and multiple groups. One of the major insights is that data generated by either multiple collectives or by coordination noise can be viewed as originating from multiple data distributions.

**Questions:**

- Could the authors please provide some concrete examples of "algorithmic collective action" within the context of machine learning?
- Could the authors elaborate further on the implications of your main results for machine learning?
- What does "Sync or Sink" in the title refer to? Could the authors clarify its meaning?

**Ethical Concerns:**

["NO or VERY MINOR ethics concerns only"]

**Final Justification:**

After reviewing the rebuttal, I am keeping my original score. While the topic itself is relevant to ML, the paper's connection to the ML community is not yet clearly stated in its submission version, which is a crucial unresolved point that may impact readability.

**Limitations:**

A potential limitation, which could also be framed as a limitation, is the paper's unclear connection to the machine learning domain. The authors should consider explicitly discussing the boundaries and the scope of their findings in relation to typical machine learning problems.

**Paper Formatting Concerns:**

No formatting concerns.

**Quality:**

2

**Strengths And Weaknesses:**

**Weaknesses:** The review's primary concern is the paper's positioning. The reviewer is not entirely certain that the work, as presented, falls within the scope of the machine learning field. The link could be made much stronger.

---

> ### Author Rebuttal · Authors · 2025-07-30
>
> We thank the reviewer for their engagement with the paper. We will restate the reviewers' concerns (grouped where appropriate) and respond inline.
>
> *The review's primary concern is the paper's positioning. The reviewer is not entirely certain that the work, as presented, falls within the scope of the machine learning field. The link could be made much stronger.*
>
> *Could the authors please provide some concrete examples of "algorithmic collective action" within the context of machine learning?*
>
> Algorithmic Collective Action is an emerging field within machine learning, where the goal is to understand the susceptibility of machine learning models to this type of coordinated behavior. It has important connections to work on strategic classification and data poisoning. Indeed much of the prior work looks at how a small set of strategically behaving points can undermine classification accuracy or lift in recommender system context. Understanding how, when, and why these data distributions occur is an essential for trustworthy machine learning.
>
> Algorithmic Collective Action has appeared in machine learning venues, including in ICML [[HMMDZ23](https://proceedings.mlr.press/v202/hardt23a.html), [BDFSS24](https://dl.acm.org/doi/10.5555/3692070.3692207)] and NeurIPS [[BMD24](https://proceedings.neurips.cc/paper_files/paper/2024/file/d79792543133425ff79513c147dc8881-Paper-Conference.pdf)]. The presence of an [Algorithmic Collective Action workshop at this year’s NeurIPS](https://acaworkshop.github.io/)  also speaks to its place within the Machine Learning community. Our work is a direct follow up to these important lines of machine learning work and has carved out an important and growing subfield.
>
> Collective action has been growing and its impacts will only grow with time. [[SHMD24](https://dl.acm.org/doi/10.1145/3706598.3713966)] has documented many cases that have both been studied by academics as well as noted in media. It has been documented in the case of [AI art](https://arstechnica.com/information-technology/2022/12/artstation-artists-stage-mass-protest-against-ai-generated-artwork/), [writing](https://www.nytimes.com/2023/07/15/technology/artificial-intelligence-models-chat-data.html) among others. Ignoring why and how these changes affect ML systems would be a disservice to the community. These types of strategic behaviors are only expected to grow, and more broadly, recognizing that data itself is, across many domains, generated by strategic actors is important to consider in how it can affect downstream systems.
>
> *Could the authors elaborate further on the implications of your main results for machine learning?*
>
> As mentioned in our introduction/prior works section, previous research has shown that a small, perfectly coordinated group, both via experiments and proven bounds, can effectively influence these ML systems [HMMDZ23]. Other work has shown empirically, however, that when two or more collectives are acting in a system, there can be conflicting outcomes [KVKS25]. We sought to address a gap to understand how sensitive is collective action to multiple distributional sources: this can either come from a lack of perfect coordination or it can come from a 2nd independent group acting on a system. We provide new bounds in the case of multiple distributions and we conduct experiments to show how noise can degrade the efficacy of collective action; prior work has only focused on the perfectly coordinated case. As mentioned previously, understanding how competing groups affect the underlying data distributions of ML models is poised to become a much larger technical and practical challenge.
>
>
> *What does "Sync or Sink" in the title refer to? Could the authors clarify its meaning?*
>
> “Sync or Sink” refers to how perilous this type of collective action may be. Traditional collective action generally requires a high degree of coordination (i.e for everyone to be in “sync” with each other), otherwise the whole project can easily fail (or “sink”). We appreciate the feedback that this was not obvious and will certainly seek to clarify this!

---

> > ### Comment · Reviewer_Kze9 · 2025-08-07
> >
> > Thank you for your detailed response and for clarifying your perspective on Algorithmic Collective Action as an emerging field.
> >
> > While I find the research direction promising, my main suggestion is to revise the manuscript to make the explicit connection to the machine learning community clearer and more accessible. I believe revisions are needed to more explicitly bridge the concepts to our community.
> >
> > Specifically, I would encourage you to:
> >
> > - Strengthen the introduction to more clearly situate the problem and its importance for ML community.
> >
> > - Weave these connections throughout the paper to ensure the relevance is clear, not just at the Intro. Some illustrative case studies might be effective here.
> >
> > Given that these revisions are needed, I will maintain my current score. For full transparency, I should note that my review confidence is 2, as I am not an expert in this specific sub-domain.
> >
> > Thank you for your work on this interesting topic.

---

> > > ### Author Response · Authors · 2025-08-07
> > >
> > > We thank the reviewer for their continued engagement! We appreciate the additional feedback and aim to incorporate it in our work.
> > >
> > > We will add further connections to the broader community, specifically expanding about our discussion about how this affects data distributions (line 27) and connections to distributional shifts. We can further will also discuss some of the examples we discuss in lines 105-113 in the context of these distributional shifts and broader connection to the overall ML community.

---

### Official Review · Reviewer_y8VU · 2025-07-05

**Clarity:** 2
**Significance:** 3
**Originality:** 3
**Rating:** 4
**Confidence:** 4

**Summary:**

This paper studies the effectiveness of algorithmic collective action in the presence of coordinated noise and multiple collectives, proposes a theoretical lower bound and conducts empirical analysis. It is found that data perturbations, whether caused by multiple collectives or uncoordinated behaviors within a collective, can be modeled as multiple data distributions, and for the first time, a lower bound on the success rate of collective action is proposed based on this perspective. The impact of noise is studied through resume classification experiments, and it is found that high noise can significantly reduce the success rate, and small-scale low-noise groups may be better than large-scale high-noise groups.

**Questions:**

My questions are:
Q1: Have the authors condered using more detailed explanations or vivid demonstrations of the problem being solved in the main part of the paper?
Q2. In the experiment, the target categories only involved high and low frequencies, and did not cover the characteristics of medium frequencies or more detailed categories. Does this paper need this consideration?

**Ethical Concerns:**

["NO or VERY MINOR ethics concerns only"]

**Final Justification:**

Although several responses from the authors were given, I still feel that some of my concerns were not well addressed and the response were not satisfactory. I am going to keey my original score.

**Limitations:**

yes

**Paper Formatting Concerns:**

no major formatting issues in this paper

**Quality:**

2

**Strengths And Weaknesses:**

Strengths: This paper unifies noise and multiple groups into a multi-distribution problem for the first time and establishes a rigorous mathematical framework. It clearly demonstrates the trade-off between organizational coordination and collective scale, and through experiments with specific noise types and strategies, reveals the differential impact of noise on different strategies. In addition, this research has clear practical value and can provide a reference for decision-making on whether to prioritize coordination of small groups or expansion of scale in the future.

Weaknesses: If the distribution caused by group behavior or noise is difficult to clearly divide in practice, it may make the distribution difficult to separate. In addition, it mainly focuses on text classification tasks, the data scale and type are relatively simple, and the experiment is only conducted on a multi-label classification set. In addition, only benign noise is considered. If it is completely interfering noise, it may have an impact on the model. Finally, the differences in the sensitivity of different learning algorithms to noise are not considered.

---

> ### Author Rebuttal · Authors · 2025-07-30
>
> We thank the reviewer for their feedback and excitement in this area of multiple distributions and multiple collectives. We will first address the weaknesses and then address the specific questions.
>
> [W1] *If the distribution caused by group behavior or noise is difficult to clearly divide in practice, it may make the distribution difficult to separate*
>
> We agree that in practice, depending on the source of the multiple distributions, it can be difficult to clearly separate out. In the case of noise related distributional differences in particular, this can be difficult to separate out in practice. Our goal with this bound is to concertize the specific factors driving decreased efficacy across multiple distributions more broadly (i.e due to either noise or multiple collectives). Multiple distributions coming from multiple distinct groups with their own objectives would be easier to divide than noise related ones. We will clarify within the paper that our bound handles both the situation of noise or multiple collectives, and in practice it may be difficult to separate out the noisy distribution from the non-noisy distribution.
>
> [W2] *In addition, it mainly focuses on text classification tasks, the data scale and type are relatively simple, and the experiment is only conducted on a multi-label classification set.*
>
> Our focus on this setting is to directly make an apples to apples comparison with [HMMDZ23, KVKS25]. Because both of these works look at this classification setting, one with the single collective bound in mind [HMMDZ23] and one with the multiple collectives setting [KVKS25], our aim was to offer the most compelling comparison to prior work. As in prior work, we expect results on this task to provide some degree of generalized insights (i.e., findings should generally hold across many domains), though agree that future experimental work should seek to explore additional distributions and tasks.
>
> [W3] *In addition, only benign noise is considered If it is completely interfering noise, it may have an impact on the model.*
>
> Our experiments were primarily inspired by potential ways that collective action might degrade due to coordination challenges. However, our bounds describe the relationship between any two distributions and hence, hence they do allow for analysis in that scenario. Direct interference of a collective’s work is an important problem, but here the focus is on how even a well intentioned group can fail even when everyone is aligned on the same goal.
>
> [W4] *Finally, the differences in the sensitivity of different learning algorithms to noise are not considered.*
>
> We primarily focus on expanding the general case [HMMDZ23], though specialization to specific learning algorithms is important in practice. Future work can expand upon these bounds to different learning algorithms.
>
> For the questions:
>
> [Q1] *Have the authors considered using more detailed explanations or vivid demonstrations of the problem being solved in the main part of the paper?*
>
> We noted several instances of this type of collective action being done in the real world within the introduction. We cut some of the details for space, but we can further expand on some of the academic case studies [SHMD24] as well as media reported instances. for example in [art](https://arstechnica.com/information-technology/2022/12/artstation-artists-stage-mass-protest-against-ai-generated-artwork/) and [writing](https://www.nytimes.com/2023/07/15/technology/artificial-intelligence-models-chat-data.html) in the introduction.
>
> [Q2] *In the experiment, the target categories only involved high and low frequencies, and did not cover the characteristics of medium frequencies or more detailed categories. Does this paper need this consideration?*
>
> We focused on high/low frequencies just to show the potential range of behaviors, more medium frequencies would lie somewhere in between. We can add those figures in for completeness.

---

> > ### Comment · Reviewer_y8VU · 2025-08-06
> >
> > Thank you for the author's reply. I still have a few minor questions about this: 1. If the population distribution and noise are not clearly distinguished, will it have a significant impact on the model's performance? Could the author provide a further explanation for the statement "even a welintentioned group can fall even when evervone is aligned on the same goal" mentioned in the reply "W2"?

---

> > > ### Author Response · Authors · 2025-08-07
> > >
> > > We thank the reviewer for continued engagement and interest in further details!
> > >
> > > *If the population distribution and noise are not clearly distinguished, will it have a significant impact on the model's performance*
> > >
> > > We will respond based on our understanding of the question, if this wasn't what the reviewer was looking for, feel free to respond saying so. We will respond with further clarification.
> > >
> > > Theorem 3 states:
> > >
> > > $S(\alpha_1) \geq 1 - \frac{\alpha_2}{\alpha_1} \mathcal{P}_2(\mathcal{X}_1) \frac{(1 - \epsilon_1)\Delta_1^2(y_1) + \epsilon_1}{1 - 2\epsilon_1} - \frac{1-\alpha}{\alpha_1} \mathcal{P}_0(\mathcal{X}_1)\frac{(1 - \epsilon_1)\Delta_1^0(y_1) + \epsilon_1}{1 - 2\epsilon_1} $
> > >
> > > In the case of noise, we have the baseline distribution as $\mathcal{P}_0$ the faithfully executing collective as $\mathcal{P}_1$ and the noisy variation distribution as $\mathcal{P}_2$,
> > >
> > > If the overlap between $\mathcal{P}_1$ and $\mathcal{P}_2$ (the perfectly coordinated collective portion and the noisy variation), is high then the $\mathcal{P}_2(X_1)\Delta_1^2(y_1)$ in the success bound would be high and thus lead to decrease effectiveness of the collective.
> > >
> > > If the overlap between $\mathcal{P}_0$ and $\mathcal{P}_2$ is high, then the $\mathcal{P}_0(X_1)\Delta_1^0(y_1)$ could indirectly be inflated. Another way is to consider it is if $\mathcal{P}_0$ and $\mathcal{P}_2$ are very similar, then in the most extreme case, the $\alpha_2$ fraction associated with the noisy distribution can be viewed as belonging to $\mathcal{P}_0$ distribution. Instead of there being  $1 - \alpha = 1-\alpha_1 - \alpha_2$). points in the $\mathcal{P}_0$ distribution, it would be $1-\alpha_1$ (representing $\alpha_2$ fraction following $\mathcal{P}_2$ being nearly similar to $\mathcal{P}_0$ and hence this term would be inflated and thus signaling reduced efficacy of the collective.
> > >
> > > If this wasn't what the reviewer was seeking clarification about please do let us know so we can add the appropriate clarification!
> > >
> > > *Well intentioned group*
> > >
> > > Stepping back, consider a group (any group for that matter) that's trying to achieve some goal. Looking to the classic collective action literature, (Olson and Ostrom) we often find collective organizing is difficult even if the outcome is one that everyone agrees upon would be good.  When we talk about ``well-intentioned" groups, our aim is to understand what types of groups can succeed, even if there's not a direct malicious actor trying to undermine them. Some groups fail because of lack of good coordination mechanism, lack of coherent strategy, etc or any of the number of failure modes described by Olson or Ostrom. We abstracted these specifically types of potential failures into this general concept of "noise" where, at a high level, any action that deviates from the collective's intended goal is viewed as 'noise' in the mathematical framing.
> > >
> > > Our bound then to manifest them in terms of considering the faithfully executing group and the noisy variant. Our goal was to see how formalize how factors such the size of the noisy group ($\alpha_1$ vs $\alpha_2$) and the overlap between them (e.g $\mathcal{P}_2(\mathcal{X}_1)$) affects the efficacy of collective action. In this view, we can start to understand what types of groups might be more successful. In Section 7.2, we briefly discuss potential tradeoffs between overall size of group and noise, postulating that there may be potential tradeoffs in increasing size vs keeping noise low. This type of characterization is important going forward in understanding what types of groups may be likely to succeed with algorithmic collective action.

---

> > > > ### Comment · Reviewer_y8VU · 2025-08-08
> > > >
> > > > Thank you for your detailed reply, author. Regarding the issue where there is no obvious difference between distribution and noise, I am more focused on how robust the model is to noise? Furthermore, does "well-intentioned groups" refer to groups where the majority of members have the same goals, or does it refer to the correctness of the members' goals?

---

> > > > > ### Author Response · Authors · 2025-08-08
> > > > >
> > > > > Thank you for the clarifying question!
> > > > >
> > > > > Regarding the susceptibility of the model to noise, we don't explicitly characterize that in this work, our characterization of the learning model is just the $\epsilon_1$ suboptimality. There is some implicit relationship in the $\Delta_1^0(y_1)$ term as it looks at the term $\mathcal{P}_0(y | x) - \mathcal{P}_0(y_1 | x)$.
> > > > >
> > > > > Other work has more specifically the role of the specific learning algorithm. In particular [BDFSS24] specifically looks at bounds on distributional robust objectives, which they are able introduce include a factor of distributionally robust in some f-divergence ball as part of their bound, relating the robustness of the underlying learning objective to susceptibility of collective action. Our focus on the $\epsilon_1$ suboptimal case but expanding this analysis to more specific classes of models/learning algorithms is an important next step. We can add this point to the discussion!
> > > > >
> > > > > For the "well-intentioned group" the way we think about it is that everyone has the same nominal notion of what they want to accomplish as a group (make my favorite artist popular, protect my artistic style, make our resumes classified a certain way etc.) They may not implement the correct action for a variety of reason (miscommunication, not good at following instructions etc.)  Many of our experimental conditions we tested looked at these ways that someone might "slip up"

---

### Decision · Program_Chairs · 2025-09-17

**Decision:**

Reject

**Comment:**

The authors consider an extension of the algorithmic collective action problem where the data generated by the collective(s) follows two distributions: one distribution designed to achieve the goal desired by the collective, and one that represents either noise or a different collective with distinct goals.  They derive theoretical lower bounds on the probability that the collective achieves its objective, depending on (a) whether the collective can modify their labels as well as input features; and (b) how the two distributions differ (cross-signal overlap, suboptimality gap and collective size).  Empirical results are presented for a resume classification dataset used in previous algorithmic collective action work, assuming various types of noise (errors in strategy implementation by the collective) and measuring the impact of noise on the success of the collective.

Strengths:
* The question of how the success of a collective is impacted by lack of coordination (either because of errors or distinct objectives) is a natural and valuable extension of prior work on algorithmic collective action.
* The paper presents formal lower bounds on the success rate of the collective and supports these with empirical studies.
* The derivation of the theory is clear, well-written, and can inform questions around tradeoffs between larger collective size and the tighter coordination that might be achieved by a smaller collective.

Weaknesses:
* Inaccessible writing style for machine learning audiences: the introduction and the related work do not specify directly the connection to ML, and rely on familiarity with prior work in the field of algorithmic collective action.
* Novelty is limited/incremental: as one reviewer notes, the theoretical contribution is a straightforward generalization of prior work, the experimental results confirm what one would expect, and concrete real-world case studies are lacking.
* The empirical study is limited to a single classifier on a single dataset, and it is not clear to what extent any takeaways from the experiments would generalize to a wider range of settings.
* It is not clear how tight the theoretical bounds are or how useful they will be in practice.

Reasons for accept/rejection decision: I believe that the amount of rewriting needed to make the paper accessible and useful to a machine learning audience is too large to accept the paper without an additional round of revision and reviewing. I also share reviewer concerns about novelty and the magnitude of the contribution, given limited empirical studies and theoretical bounds that seem to be a straightforward extension of the previous paper's proofs.  Thus I believe that the paper has potential but is not ready for acceptance at its present stage.

Discussion: There was a robust author-reviewer discussion addressing many of the reviewers' concerns including the connection of the work to the machine learning field, justification of the modeling of noise and/or adversaries as multiple distributions, and the difference between strategy and signal.  One major criticism was resolved, as the reviewers reached consensus that algorithmic collective action is "in scope" for the machine learning field.  However, all reviewers remained concerned with the clarity of the exposition and its accessibility to the machine learning community, as the algorithmic collective action problem and some of its important facets (e.g., strategy vs. signal) were not well described, and the introductory part of the paper would require substantial rewriting.  Additional concerns remained about limited/incremental novelty of the work, and how empirical results generalize across different classifiers with different robustness to noise.  Overall, while three of the four reviewers rated the paper "borderline accept" to one "borderline reject" (Kze9), reviewers did not feel that their concerns were adequately addressed (y8VU), that the impact and contribution were enough (yFBj), or that they could trust the authors to make the paper sufficiently accessible to a machine learning audience (BN7G).